# Two roles for the yeast transcription coactivator SAGA and a set of genes redundantly regulated by TFIID and SAGA

**Rafal Donczew†, Linda Warfield†, Derek Pacheco, Ariel Erijman‡, Steven Hahn***

Fred Hutchinson Cancer Research Center, Seattle, United States

**Abstract** Deletions within genes coding for subunits of the transcription coactivator SAGA caused strong genome-wide defects in transcription and SAGA-mediated chromatin modifications. In contrast, rapid SAGA depletion produced only modest transcription defects at 13% of protein-coding genes – genes that are generally more sensitive to rapid TFIID depletion. However, transcription of these 'coactivator-redundant' genes is strongly affected by rapid depletion of both factors, showing the overlapping functions of TFIID and SAGA at this gene set. We suggest that this overlapping function is linked to TBP-DNA recruitment. The remaining 87% of expressed genes that we term 'TFIID-dependent' are highly sensitive to rapid TFIID depletion and insensitive to rapid SAGA depletion. Genome-wide mapping of SAGA and TFIID found binding of both factors at many genes independent of gene class. Promoter analysis suggests that the distinction between the gene classes is due to multiple components rather than any single regulatory factor or promoter sequence motif.

**\*For correspondence:**
shahn@fredhutch.org

†These authors contributed equally to this work

**Present address:** ‡New England Biolabs, Ipswich, United States

**Competing interests:** The authors declare that no competing interests exist.

## Introduction

Efficient in vitro transcription by RNA Polymerase II (Pol II) is initiated with six basal transcription factors (TATA binding protein (TBP), TFIIA, TFIIB, TFIIE, TFIIF, TFIIH), Pol II enzyme and TATA-containing promoter DNA. Cellular mRNA transcription, much of which initiates at non-TATA-containing promoters, also requires one or more coactivator complexes such as Mediator, TFIID, SAGA, NuA4 and Swi/Snf, that either directly contact the basal transcription machinery and/or modify chromatin (*Allen and Taatjes, 2015*; *Doyon and Côté, 2004*; *Hahn and Young, 2011*; *Hantsche and Cramer, 2017*; *Nogales et al., 2017*; *Rando and Winston, 2012*; *Sainsbury et al., 2015*). Often recruited by transcription activators, coactivators play roles in processes such as promoter recognition, assembly of the basal factors into a functional preinitiation complex (PIC), catalysis of covalent chromatin modifications, nucleosome positioning, and stimulation of early steps in the transition from transcription initiation to elongation. Although we know much about the structure, enzymatic functions, and protein-protein interactions among many of the above factors, relatively little is known about mechanisms that determine genome-wide coactivator specificity and the molecular mechanisms of how they promote induction and maintenance of activated transcription.

The yeast coactivators SAGA and TFIID share five subunits (*Grant et al., 1998*) but are otherwise unrelated. SAGA contains four known activities: activator binding, histone H3-acetylation, histone H2B deubiquitylation and TBP binding (*Han et al., 2014*; *Helmlinger et al., 2011*; *Lee et al., 2011*; *Liu et al., 2019*; *Setiaputra et al., 2015*). In addition to performing chromatin modifications, SAGA was recently shown to function as a TBP-DNA loading factor, mediated in part via TFIIA and its TBP-binding subunit Spt3 (*Papai et al., 2019*). TFIID is a large dynamic complex, comprised of TBP and 13–14 Taf (TBP associated factor) subunits, that functions in activator binding, promoter recognition

and PIC formation (*Bieniossek et al., 2013*; *Hahn and Young, 2011*; *Nogales et al., 2017*). Prior studies showed that TFIID can bind one or more short downstream metazoan promoter elements and that it undergoes large conformational changes while positioning TBP on DNA (*Kolesnikova et al., 2018*; *Louder et al., 2016*). CryoEM structures of human TFIID and TFIIA-TFIID-DNA revealed that this complex seems incompatible with PIC formation (*Patel et al., 2018*). Several Tafs in the TFIID-DNA complex are in position to clash with TFIIF and Pol II in the PIC and the DNA bend in the TFIIA-TFIID-DNA complex is different from that in the closed state PIC. From this it was proposed that TFIID, in conjunction with TFIIA, functions as a TBP-loading factor and that the Tafs must either dissociate or undergo major rearrangement before PIC assembly (*Joo et al., 2017*; *Patel et al., 2018*).

The genome-wide promoter specificities of SAGA and TFIID have been the subject of much investigation. Early work showed that at least some yeast genes can be activated and transcribed after depletion or inactivation of TFIID (Taf) subunits (*Kuras, 2000*; *Li et al., 2000*; *Moqtaderi et al., 1996*; *Walker et al., 1996*). Pioneering genome-wide studies later suggested that both TFIID and SAGA contribute to steady state mRNA expression of most yeast genes, but that transcription of nearly all genes is dominated by the dependence on either TFIID or SAGA (*Huisinga and Pugh, 2004*). From this study, the 'TFIID-dominated' class represented ~ 90% of Pol II-transcribed genes, many of which lacked a consensus TATA element, while the 'SAGA-dominated' genes are enriched for TATA-containing and stress-inducible genes. Formaldehyde crosslinking methods showed that promoters from the TFIID-dominated class generally have high levels of TFIID-subunit crosslinking while lower levels of crosslinking are often observed at the SAGA-dominated genes (*Kuras, 2000*; *Li et al., 2000*; *Rhee and Pugh, 2012*).

Several subsequent findings challenged these initial models. First, mutations affecting genome-wide transcription are now known to alter mRNA stability, making prior steady state mRNA measurements used in earlier studies of TFIID and SAGA specificity problematic (*Haimovich et al., 2013*; *Munchel et al., 2011*; *Sun et al., 2012*). Second, mapping SAGA-dependent chromatin modifications showed that SAGA modifies chromatin at nearly all expressed genes, rather than only at the SAGA-dominated subset (*Bonnet et al., 2014*). Third, formaldehyde-independent mapping of SAGA and TFIID using the MNase-based ChEC-seq approach identified TFIID and SAGA binding at both gene classes (*Baptista et al., 2017*; *Grünberg et al., 2016*). Finally, native Pol II ChIP and/or analysis of newly synthesized mRNA found that the majority of genes showed decreased transcription upon deletion or depletion of SAGA or TFIID subunits (*Baptista et al., 2017*; *Warfield et al., 2017*). However, the proposal that TFIID is generally required for transcription was recently disputed based on anchor-away depletion of the TFIID subunit Taf1 (*Petrenko et al., 2019*). Analysis of the top 10% of transcribed genes, analyzed by Pol II-ChIP, showed that many genes in the SAGA-dominated class were only modestly sensitive to Taf1 depletion.

We have now investigated the genome-wide specificities of TFIID and SAGA by monitoring levels of newly synthesized RNA after rapid TFIID and SAGA depletion. This approach has greater sensitivity and lower background than methods used in earlier studies and allows reliable quantitation of mRNAs for ≥ 83% of yeast genes after factor depletion. We find that SAGA has two separable functions in gene expression: (1) a general function important for transcription of nearly all genes. This function was revealed by deletions in genes encoding key SAGA subunits. (2) a gene-specific role revealed by rapid depletion of SAGA function, where only 13% of genes are modestly affected. Surprisingly, this smaller gene set is, on average, more sensitive to rapid TFIID depletion. These and other results suggest that TFIID and SAGA function is substantially redundant at this set of co-regulated genes that we term 'coactivator-redundant' rather than 'SAGA-dominated' or 'Taf-independent' as previously proposed. The other 87% of yeast genes that we term 'TFIID-dependent' are very sensitive to rapid TFIID depletion but show little or no change upon rapid SAGA depletion. We further found that the role of SAGA in global transcription is strongly linked to SAGA-dependent chromatin modifications, particularly Gcn5-dependent H3 acetylation. This mark, that we find is required for efficient global transcription, is significantly reduced in the *SPT* deletions but declines only slowly in the rapid depletion experiments. Finally, we examine features that distinguish the two gene sets and the role of SAGA in maintenance of ongoing activated transcription at the coactivator-redundant genes.

## Results

### SAGA has two separate roles in transcription

To investigate the basis for conflicting reports on the genome-wide roles of SAGA and TFIID, we used 4-thioU RNA-seq to quantitate transcriptional changes that are caused by various SAGA depletion strategies. Yeast were grown in synthetic or rich glucose media and RNAs labeled for 5 min, followed by purification of the labeled RNA and quantitation by sequencing. This method has much higher sensitivity and lower background compared with Pol II ChIP and allows reproducible quantitation of newly-synthesized mRNA from ≥83% of Pol II-transcribed genes (Materials and methods). Experiments, except where noted, were performed in triplicate and there was low variation across the set of analyzed genes (*Figure 1—figure supplement 1*).

We examined genome-wide transcription changes caused by deletions in the SAGA subunits: *spt3Δ*, *spt7Δ*, or *spt20Δ* or by rapid depletion of several SAGA subunits using the auxin-degron system (*Nishimura et al., 2009*). To increase the probability of efficient SAGA inactivation, double degron strains permitted simultaneous depletion of two SAGA subunits, either Spt3/7 or Spt3/20. Spt3 is important for TBP binding and DNA loading while Spt7 and Spt20 are important for the integrity of SAGA (*Han et al., 2014*; *Lee et al., 2011*; *Liu et al., 2019*; *Papai et al., 2019*; *Sterner et al., 1999*). The depletion by degron approach was also applied to the individual TFIID subunits Taf1, Taf7 and Taf13 that are located in two different TFIID lobes (*Patel et al., 2018*; *Warfield et al., 2017*). Protein degradation was induced by adding the auxin indole-3-acetic acid (IAA) for 30 min, after which time, ≤10% of protein remained (*Figure 1—figure supplement 2*). In the degron strains, mRNA levels from nearly all genes in the absence of IAA was within ~ 20% of that from WT strains lacking a degron (*Figure 1—figure supplement 3*) and addition of IAA to a WT strain had no effect on gene expression (see below).

We found a striking difference in genome-wide transcription defects when comparing rapid inactivation of SAGA, a strain containing an *SPT3* deletion, or rapid inactivation of Taf13 (*Figure 1A*, *Supplementary files 1*, *2*). For example, the *spt3Δ* strain showed strong genome-wide transcription defects for nearly all genes while degron-induced inactivation of Spt3/Spt20 for 30 min caused only modest defects at a small subset of genes. The strong decrease in genome-wide transcription in the *SPT* deletion strains is in good agreement with prior results obtained using dynamic transcriptome analysis (DTA) and 4-thioU-labeled RNA (*Baptista et al., 2017*). Rapid depletion of Taf13, a key TBP-interacting subunit located in TFIID lobe A, caused strong genome-wide transcription defects at most genes as previously observed using native Pol II ChIP (*Warfield et al., 2017*). For all strains assayed, the changes in gene expression showed poor correlation with a stress-response transcription signature (*O'Duibhir et al., 2014*), indicating that the changes we observed are not due to a general stress response (*Figure 1—figure supplement 4A*). The degron-containing strains showed ≥85% viability after 30 min of IAA treatment (*Figure 1—figure supplement 4B*) (*Warfield et al., 2017*).

To determine whether subsets of genes are differentially affected by any of these depletion strategies, the $\log_2$ change in transcription values from the SAGA deletion strains and TFIID-degron strains were clustered by k-means algorithm with the results shown in *Figure 1B* and *Supplementary file 3*. Grouping into two clusters showed that the two sets of genes respond differently to TFIID or SAGA depletion. The clustering was very robust as we obtained > 95% overlap in gene categories when we clustered using results from the SAGA-degron strains instead of the *SPT* deletion strains. The largest cluster, we term 'TFIID-dependent genes' (see reasoning below), contains 87% of analyzed genes (4245 genes) and is most sensitive to TFIID depletion. These genes show a ~ 4.5 fold average decrease in transcription upon depletion of ether Taf13, Taf7 or Taf1 (*Figure 2A*, orange boxes, lanes 8–11). Results for the Taf13-degron are very similar when cells are grown in either synthetic complete (SC) or rich (YPD) glucose media with excellent correlations between Taf1, 7 and 13 depletions (*Figure 1—figure supplement 4C*). This large gene set was also sensitive to gene deletions in SAGA subunits with genome-wide transcription decreased by an average of ~2.7 fold in the *spt3Δ* strain with lesser but significant defects observed in the *spt7Δ* and *spt20Δ* strains (*Figure 2A*, orange boxes, lanes 5–7). In contrast, expression of nearly all genes in the 'TFIID-dependent' set showed little or no change upon rapid SAGA subunit depletion using the Spt3/20 and Spt3/7 degron strains (*Figure 2A*, orange boxes, lanes 2–4).

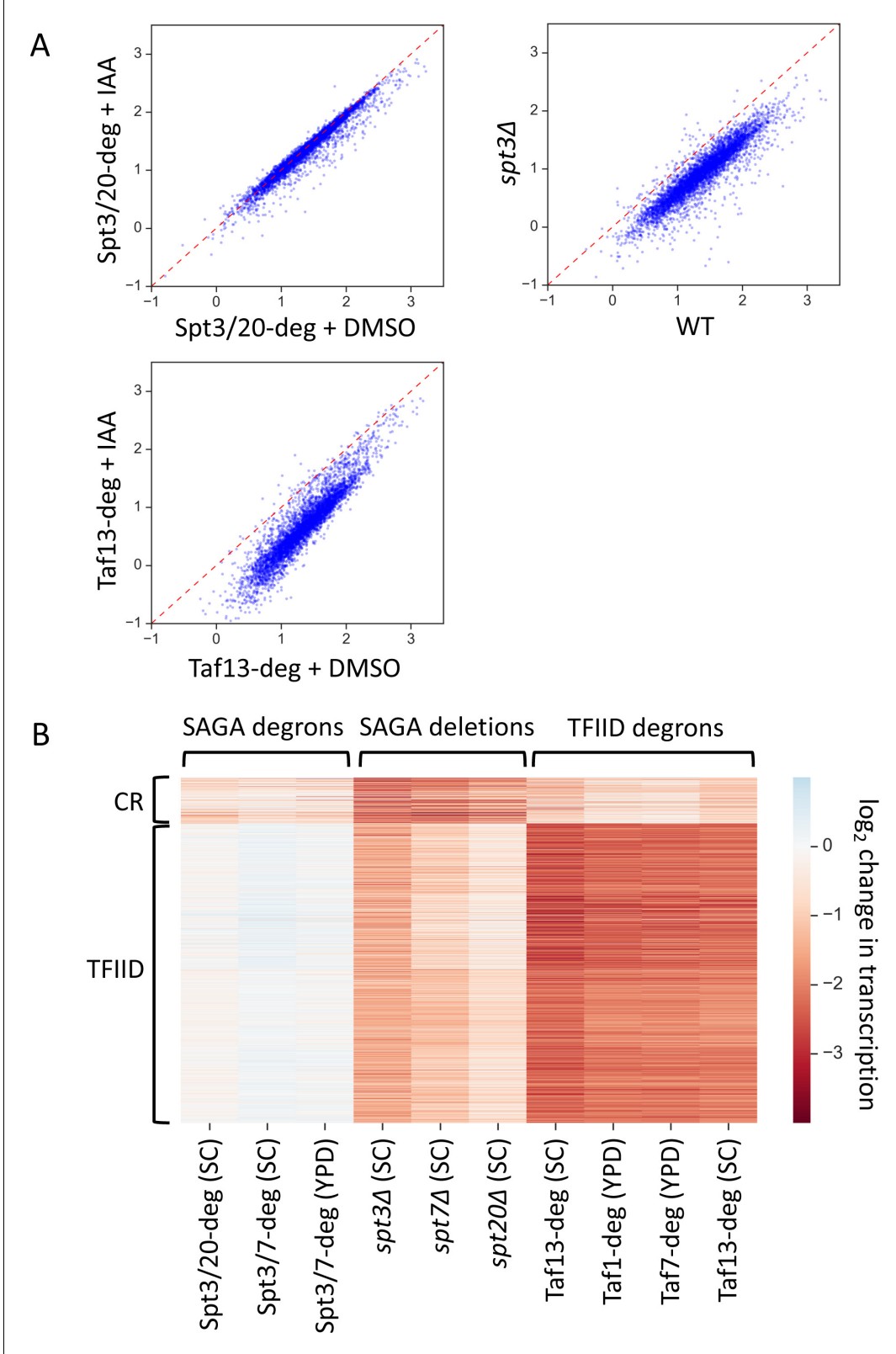

**Figure 1.** Two distinct classes of yeast genes based on SAGA and TFIID dependence. (**A**) Scatter plots comparing normalized signal per gene in corresponding samples. $Log_{10}$ scale is used for X and Y axes. Mean values from replicate experiments are plotted. All presented experiments were done in SC media. (**B**) Heatmap representation of $log_2$ change in transcription values. Mean values from replicate experiments are plotted. Genes are grouped by results of k-means clustering analysis of SAGA deletion and TFIID degron experiments. Two clusters were found to give the best separation

*Figure 1 continued on next page*

*Figure 1 continued*

using silhouette analysis. Log$_2$ change values from relevant experiments for all 4900 genes were used as an input for k-means algorithm ('KMeans' function from Python sklearn.cluster library with default settings).

The online version of this article includes the following source data and figure supplement(s) for figure 1:

**Figure supplement 1.** Biological replicates for 4-thioU RNA-seq experiments show low variability.

**Figure supplement 1—source data 1.** Coefficient of variation values for different experiments used to plot *Figure 1—figure supplement 1*.

**Figure supplement 2.** Auxin-degron system allows for efficient degradation of target TFIID and SAGA subunits.

**Figure supplement 3.** Validation of mutant strain fitness and enrichment of nascent transcripts in 4thioU RNA-seq experiments.

**Figure supplement 3—source data 1.** Mean values of log$_{10}$ expression per gene for selected DMSO treated samples used to plot *Figure 1—figure supplement 3A*.

**Figure supplement 4.** Mutant genotype and IAA treatment for 30 minutes do not compromise viability of tested strains.

**Figure supplement 4—source data 1.** Average log$_2$ changes in transcription in the degron and deletion strains same as in *Supplementary file 3* with additional column (SGS) containing the values of slow growth signature defined in *O'Duibhir et al. (2014)*.

The smaller gene set, that we term 'TFIID/SAGA-redundant' (or 'coactivator-redundant' (CR); see reasoning below), contains 13% of analyzed genes (655 genes) and these genes are more sensitive to deletions in SAGA subunits (~3–4 fold decrease) and less sensitive to TFIID depletion (~1.5–2-fold decrease) compared with the TFIID-dependent genes. (*Figure 2A*, blue boxes, lanes 5–11). Surprisingly, these coactivator-redundant genes showed only modest transcription defects upon rapid SAGA depletion (*Figure 2A*, blue boxes, lanes 2–7). The strongest average defect was caused by the Spt3/20 degron, with an average transcription decrease of ~1.6 fold. Combined, our results show that there is a fundamental difference between cell growth for many generations in the absence of SAGA vs rapid depletion of SAGA function. Surprisingly, comparison of transcription defects caused by rapid TFIID depletion vs rapid SAGA depletion showed that the coactivator-redundant genes are, on average, more sensitive to rapid depletion of TFIID (*Figure 2A*, blue boxes, lanes 2–4; 8–11). For example, we observed a ~ 2 fold average decrease in transcription in the Taf13-degron vs ~ 1.6 fold decrease in the Spt3/20 degron.

The results above show that transcription of the smaller gene set is not dominated by either coactivator and suggests that TFIID and SAGA function is at least partially redundant at these genes. To quantitate the extent of TFIID and SAGA redundancy at all genes, we rapidly depleted both TFIID and SAGA using Taf13/Spt3 and Taf13/Spt7 double degron strains. We found that simultaneous TFIID and SAGA depletion causes a severe defect in the coactivator-redundant gene set with an average decrease of ~5.7 fold (*Figure 2B*, blue boxes, lanes 4–5; *Figure 2C*, *Figure 2—figure supplement 1A* left panel, and *Supplementary file 4*). For nearly all these genes, the transcription defect is greater than simply the sum of the defects observed in the Taf13-degron and Spt3/20-degron (*Figure 2—figure supplement 1B*), showing that there is substantial redundancy in the ability of TFIID and SAGA to promote transcription of these genes. In contrast, the TFIID-dependent genes show, on average, no additional defects in the double degron strains compared with the Taf13-degron (*Figure 2B*, orange boxes, lanes 3–5; *Figure 2—figure supplement 1A* right panel, and *Supplementary file 4*).

After 30 min of IAA treatment, the levels of each degron-tagged Spt subunit in the double degron strain are reduced ~ 90%, but it is possible that low levels of intact SAGA contribute to genome-wide transcription. However, results from the Spt3/Taf13 and Spt7/Taf13 double degron strains strongly suggests that depletion of only one SAGA subunit is sufficient to inactivate SAGA function (*Figure 2B*). This result shows that the weak effects of the SAGA degrons in otherwise WT cells is due to substantial redundancy with TFIID function at the coactivator-redundant genes rather than incomplete depletion.

## Differences between 'coactivator-redundant' genes and 'SAGA-dominated' genes

Our TFIID-dependent and coactivator-redundant gene sets overlap with the categories previously defined: 'TFIID-dominated' and 'SAGA-dominated' (*Huisinga and Pugh, 2004*), however, there are significant differences. Our coactivator-redundant genes (655 genes) contain 2/3 of the SAGA-dominated gene set (273 genes) plus an additional 311 genes that were originally characterized as TFIID-dominated (*Figure 3—figure supplement 1A*). Therefore, our coactivator-redundant gene set is

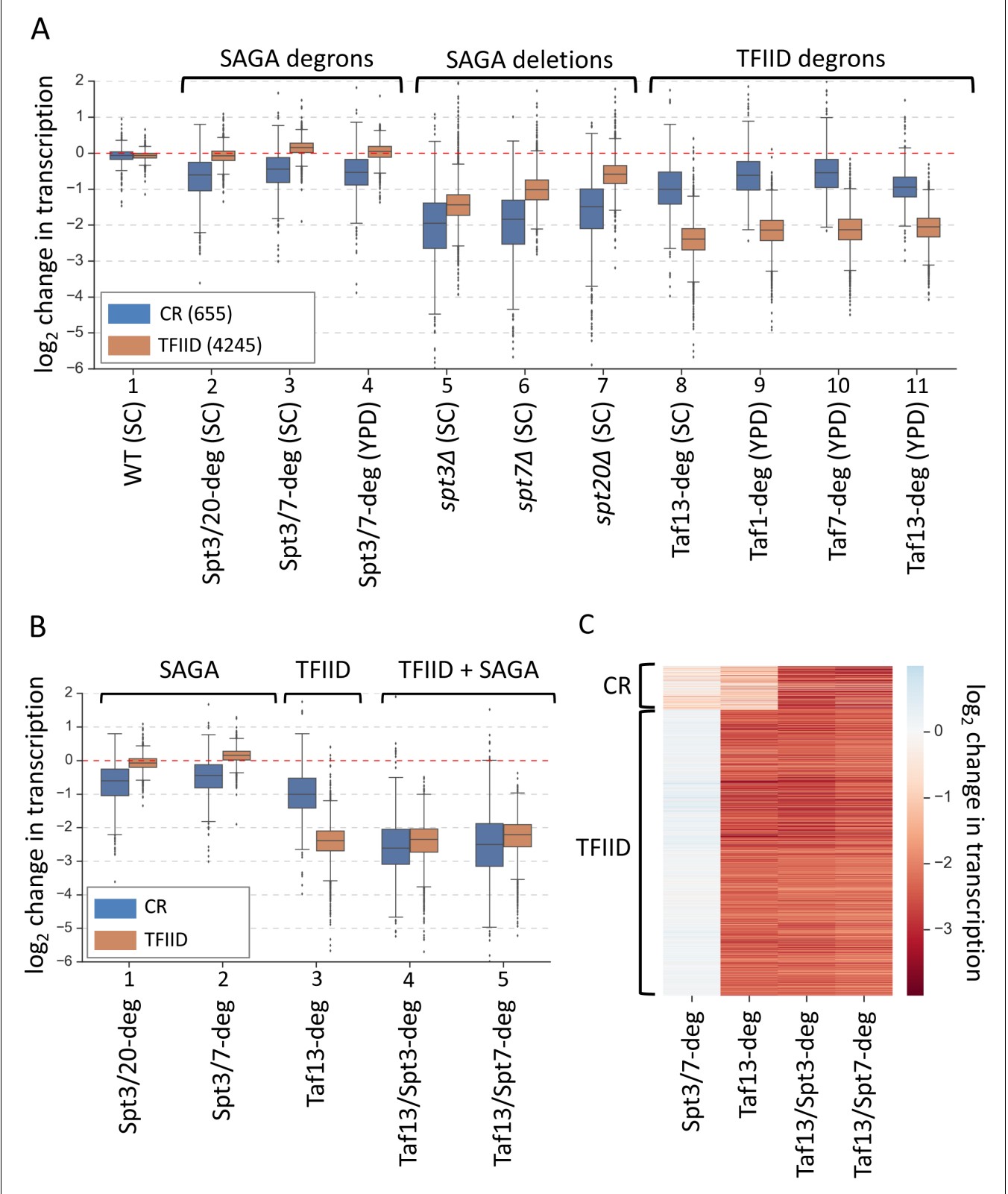

**Figure 2.** Simultaneous depletion of SAGA and TFIID severely affects transcription from almost all genes. (**A**) Box plot showing the log$_2$ change in transcription per gene upon perturbing cellular levels of SAGA or TFIID. Genes are divided into two classes according to results of the k-means clustering analysis of SAGA deletion and TFIID degron experiments from *Figure 1*. Mean values from replicate experiments are plotted. (**B**) Box plot

*Figure 2 continued on next page*

*Figure 2 continued*

and (C) heat map comparing results of depletion of SAGA or TFIID with simultaneous depletion of both coactivators. Genes are grouped into two categories from *Figure 1*. All experiments in panels B and C were done in SC media. Mean values from replicate experiments are plotted.

The online version of this article includes the following figure supplement(s) for figure 2:

**Figure supplement 1.** Coactivator-redundant genes show extensive loss transcription upon depletion of both TFIID and SAGA.

over 50% different from the SAGA-dominated gene set. There are corresponding differences in our TFIID-dependent vs the TFIID-dominated gene set. We suspect that these differences are mainly due to technical issues such as measurement of newly-synthesized vs steady state mRNA and the greater sensitivity and depth of the 4-thioU RNA-seq data compared with the microarray data used in the original analysis. These new results differ from our prior Pol II ChIP experiments that did not detect gene-specific Taf requirements (*Warfield et al., 2017*). However, it was recently suggested that this may be due to background issues with Pol II ChIP data (e.g., stalled or paused Pols) and that Pol II ChIP is best used for quantitation of changes in highly expressed genes (*Petrenko et al., 2019*). This explanation is consistent with our new findings.

Importantly, our reclassification of many genes affects the interpretation of published studies examining coactivator function and specificity that relied on the SAGA-dominated and TFIID-dominated gene lists. For example, our prior collaborative study measured gene expression defects upon rapid depletion of SAGA subunit Spt7 by anchor away (*Baptista et al., 2017*). These results showed that expression of genes in both the SAGA-dominated and TFIID-dominated gene categories are sensitive to rapid SAGA inactivation, in apparent contrast to the above results. However, comparing the specific genes tested in this earlier work with our revised gene categories shows that 4/6 of the TFIID-dominated genes assayed by Spt7 anchor away are in fact in the new coactivator-redundant gene class and are expected to be sensitive to rapid SAGA depletion. Therefore, the Spt7 anchor away results from the earlier study are largely consistent with our current findings.

## Features of the TFIID-dependent and coactivator-redundant gene classes

When genes are sorted by transcription levels, determined by the amount of 4-thioU incorporated in 5 min, the ratio of coactivator-redundant to TFIID-dependent genes is relatively similar across the range of transcribed genes except for the most highly expressed, where the coactivator-redundant class is enriched (*Figure 3A*, *Figure 3—figure supplement 1B*). The bottom four expression quintiles have a relatively constant ratio of ~10%, but the fraction of genes in the coactivator-redundant category increases to ~30% in the top quintile and ~44% in the top 10% of expressed genes. The fraction of these genes increases even more dramatically when examining only the most highly expressed genes. Yeast Pol II transcription is dominated by a small number of very highly transcribed genes: the top 2% of expressed genes accounts for ~22% of total transcription (*Figure 3B*; *Figure 3—figure supplement 1B*; *Supplementary files 1*, *2*). As an example of the coactivator bias in these exceptional genes, 47 of the top 50 expressed genes (94%; expression >195 in *Figure 3B*), are in the coactivator-redundant category (*Supplementary file 3*).

Since highly expressed genes are enriched for the coactivator-redundant class, we asked whether the sensitivity of individual genes in this set to rapid TFIID or SAGA depletion varies with expression. For example, are highly expressed genes more sensitive to rapid SAGA or TFIID depletion? *Figure 3C* shows a plot of the difference in sensitivity to rapid Taf13 or Spt3/7 depletion vs gene expression rank. From this plot, it is apparent that most of the coactivator-redundant genes (68%) are more sensitive to Taf13-depletion compared with rapid SAGA depletion (mean difference −0.49). This difference changes slightly when considering only the top 100 expressed genes that are equally divided between stronger TFIID and stronger SAGA-dependence (mean difference −0.07). Combined, our analysis shows that the relative reliance of the coactivator-redundant genes on either TFIID or SAGA is not a barrier to very high levels of transcription.

It was previously noted that the 'SAGA-dominated' genes are enriched for TATA-containing promoters (*Huisinga and Pugh, 2004*), and the gene classes 'SAGA-dominated' and 'TATA-containing' are often equated in the literature (TATA was defined as TATAWAWR; *Basehoar et al., 2004*). To investigate this, we used the list of TATA-like elements defined in *Rhee and Pugh (2012)* obtained

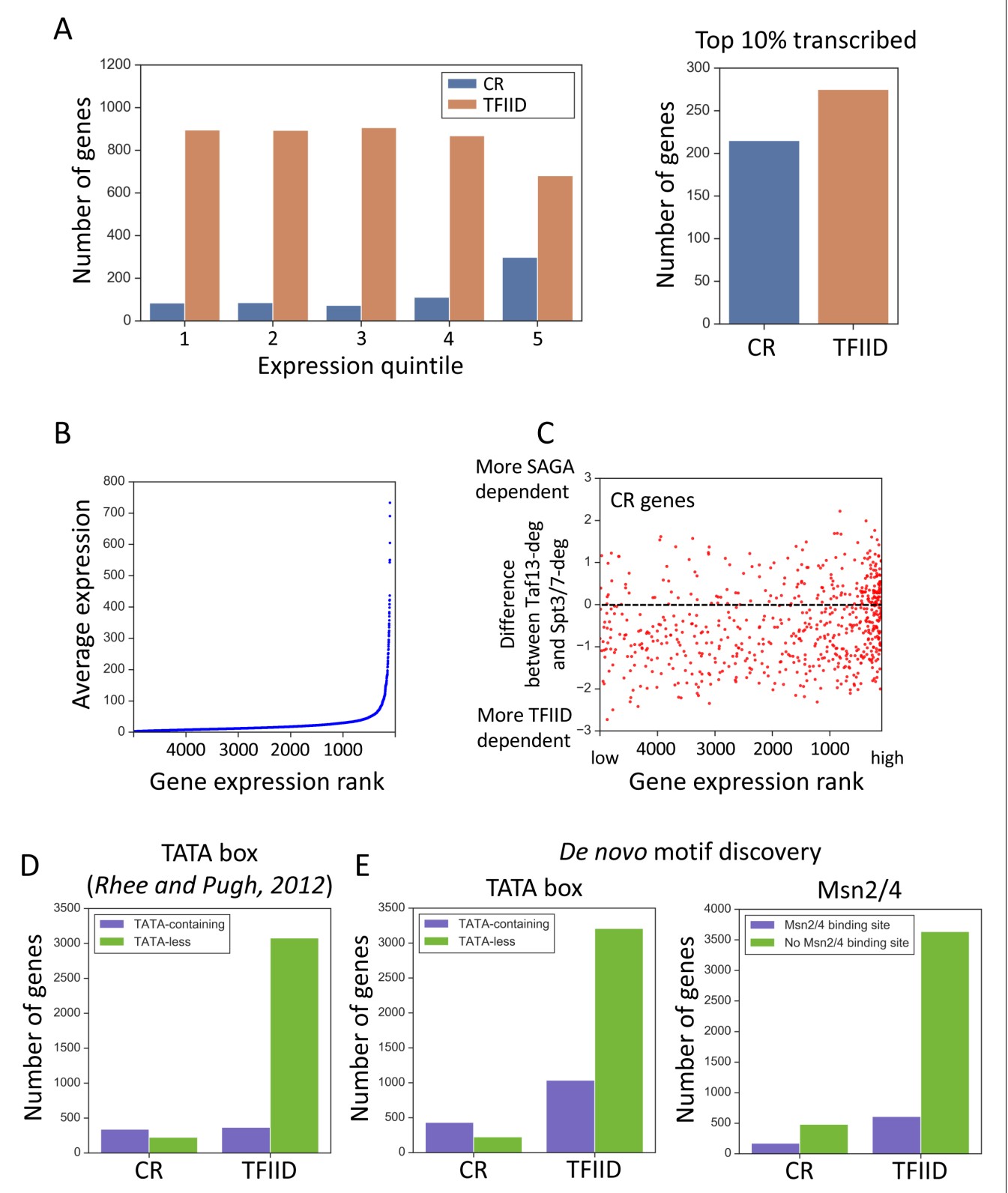

**Figure 3.** Properties associated with TFIID-dependent and coactivator-redundant genes. (**A**) Bar plot comparing the number of genes in each gene class as a function of expression level (left panel) and among the top 10% highly transcribed genes (right panel). Quintile five contains the most highly expressed genes. (**B**) Scatter plot of average expression vs the gene expression rank. Gene with rank one is most highly expressed. (**C**) Scatter plot comparing gene expression rank with the difference in sensitivity of genes to rapid Taf13 and Spt3/7 depletion. Only coactivator-redundant (CR) genes

*Figure 3 continued on next page*

*Figure 3 continued*

are plotted (655 genes). (D) Number of TFIID and coactivator-redundant (CR) genes among TATA-less and TATA-containing categories. Consensus TATA was defined as TATAWAW and the TATA-element positions from Pugh and colleagues (*Rhee and Pugh, 2012*) were used. (E) Results of a motif search guided by results of de novo motif discovery (Materials and methods). TATA box and Msn2/4 binding site were the most highly represented motifs for CR genes (*Supplementary file 5*). Each motif was searched among all 4900 promoters classified in this study. For TATA box, a consensus TATAWAW was used and the search was limited to the area from 200 bp upstream to TSS. For Msn2/4 a consensus (A/C/G)AGGGG was used (*Stewart-Ornstein et al., 2013*), and the search was limited to a region from 300 bp upstream to 50 bp upstream relative to TSS. All promoters carrying at least one consensus sequence in the defined range were classified as containing either TATA or the Msn2/4 binding site.

The online version of this article includes the following source data and figure supplement(s) for figure 3:

**Source data 1.** Table containing gene classification defined in this work and the TATA-containing/TATA-less definitions from *Rhee and Pugh (2012)*.
**Source data 2.** Tables containing gene classification defined in this work and the classification of genes into categories based on the presence of a TATA-box or Msn2/4 binding site based on promoter search performed in this work.
**Figure supplement 1.** Properties of the TFIID and coactivator-redundant gene classes.
**Figure supplement 1—source data 1.** Table containing gene classification defined in this work and the gene classes defined in *Huisinga and Pugh (2004)*.
**Figure supplement 2.** The number of ribosomal protein (RP) genes within each expression quintile.

from the YeasTSS database (www.yeastss.org), extracted the sequences of all annotated TATA-elements and reclassified promoters as TATA-containing/TATA-less based on the less stringent seven nt consensus (TATAWAW). Although TATA-containing promoters are enriched, the coactivator-redundant class is clearly not equivalent to TATA-containing genes (*Figure 3D*). Our analysis shows that 60% of coactivator-redundant genes are classified as TATA-containing but that both the coactivator-redundant and TFIID-dependent gene sets contain a nearly equal number of TATA-containing genes using the above criteria.

To further investigate differences between the gene classes, we performed de novo motif discovery for sequences surrounding the TSS of genes in each class (−400 -> +100; Materials and methods). This analysis found that the binding motifs for 14 transcription factors are preferentially enriched in one of the two gene classes (*Supplementary file 5*). For example, the binding motif for the general regulatory factor Reb1 is found near 18% of the TFIID-dependent genes vs 6% of all other genes while TATA, Msn2 and other motifs are preferentially enriched near the coactivator-redundant genes vs all other genes.

To examine these findings in more detail, we directly searched the set of genes used in our transcription analysis (~83% of all genes) for the Msn2/4 and TBP consensus binding motif (*Figure 3E*, *Supplementary file 5*; Materials and methods). This search strategy doubled the number of genes identified as containing a TBP consensus binding site (TATAWAW) and many of these genes are in the TFIID-dependent category. However, the coactivator-redundant genes are clearly enriched for TATA (65% contain this motif) while the TATA motif is found in only ~ 1/4 of the TFIID-dependent genes. We also found that the Msn2/4 motif is enriched in the coactivator-redundant genes compared with the TFIID-dependent genes, although over three times the number of TFIID-dependent genes contain Msn2/4 motifs compared with coactivator-redundant genes (173 vs 609). Since no frequently used motif is found exclusively at either the TFIID or coactivator-redundant genes, our findings suggest that there is no one transcription factor or promoter sequence motif that determines whether a gene is in the coactivator-redundant or TFIID-dependent gene class.

The 136 ribosomal protein (RP) genes have sometimes been analyzed as a distinct category of highly expressed and especially TFIID-dependent genes. From analysis of our new data, we did not find that these genes are obviously distinct from many other genes based on TFIID or SAGA-dependence. Although 95% of the RP genes belong to the TFIID-dependent category they show comparable transcription defects upon TFIID depletion as the rest of the TFIID-dependent genes. However, RP genes are clearly more highly expressed than the average gene, with 83% of the RP genes found in expression quintile 5 (*Figure 3—figure supplement 2*). K-means clustering, based on TFIID and SAGA-dependence with up to six clusters, did not segregate the RP genes into a distinct category. Alternative methods we applied (hierarchical and spectral clustering) also did not separate RP genes into a distinct category. This analysis shows that RP genes have coactivator-dependence similar to many other genes. There can be good biological reasons to examine properties of RP genes as a separate category since they are regulated coordinately in response to several signaling pathways

and this regulation is key for proper cellular growth and stress response. However, they do not seem exceptional in regard to TFIID and SAGA-dependence.

## SAGA-dependent chromatin modifications change slowly after rapid SAGA depletion

We next investigated the basis for the difference in transcription phenotypes caused by the SAGA deletion vs degron strains. SAGA functions through direct protein-protein interactions to assist with TBP recruitment and PIC formation as well as via chromatin changes mediated by Gcn5 and Ubp8 (*Bhaumik and Green, 2001*; *Dudley et al., 1999*; *Koutelou et al., 2010*; *Laprade et al., 2007*; *Larschan and Winston, 2001*; *Mohibullah and Hahn, 2008*; *Papai et al., 2019*; *Rodrí-guez-Navarro, 2009*; *Sermwittayawong and Tan, 2006*). One possibility is that the modest transcription defects caused by rapid depletion are primarily mediated by disruption of TBP-DNA loading and other non-enzymatic functions, while the stronger long-term effects of the *SPT* deletions are mediated in part through changes in chromatin modifications.

As an initial test of this model, we compared transcription levels in strains lacking specific SAGA-linked enzymatic functions (*gcn5Δ* or *ubp8Δ*), in an *spt3Δ* strain defective for SAGA TBP-binding function and, in cells after rapid SAGA-depletion (Spt3/7-degron) (*Figure 4*) (*Supplementary file 6*). We found that elimination of HAT function (*gcn5Δ*) led to a genome-wide transcription defect in both gene classes that was nearly as strong as that found in the *spt3Δ* mutant. Elimination of deubi-quitination function (*ubp8Δ*) led to more modest genome-wide defects in transcription for both gene classes.

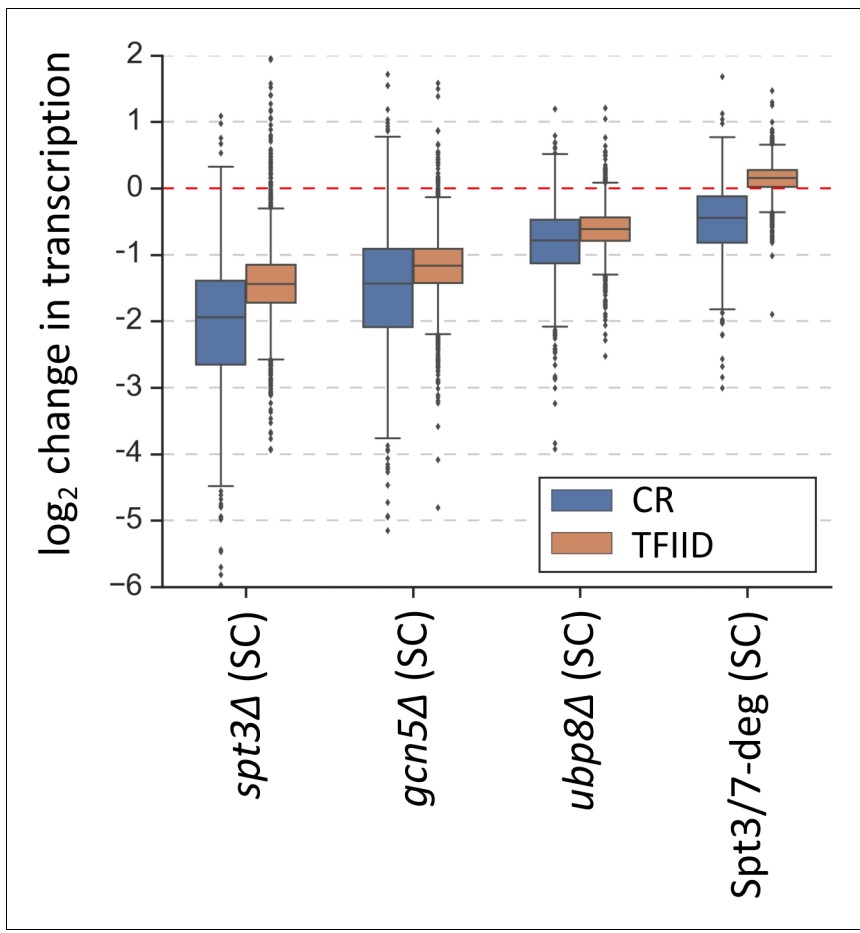

**Figure 4.** Changes in transcription (from 4thioU RNA-seq) observed in the SAGA deletion strains and in the Spt3/7 degron strain after rapid depletion. Mean values from replicate experiments are plotted and the genes are divided into TFIID-dependent and coactivator-redundant (CR) categories.

Since transcription defects in *SPT* deletion mutants are stronger than that caused by rapid SAGA depletion, we next examined the kinetics of changes in SAGA-dependent chromatin marks in the Spt-degron strains. The SAGA-dependent chromatin marks H3K18-Ac and H2B-Ub (*Daniel et al., 2004*; *Henry et al., 2003*; *Morris et al., 2007*) were monitored in the *SPT* deletion strains and after IAA addition to the Spt3/7 and Spt3/20 degron strains. Western blot analyses of whole cell extracts showed that the *spt3Δ*, *spt20Δ* and *spt7Δ* strains all had lower total levels of H3K18-Ac, ranging from ~ 34% of normal in the *spt7Δ* and *spt20Δ* strains to 70% of normal in the *spt3Δ* strain (*Figure 5A*, left panel; *Figure 5—figure supplement 1A*). H2B-Ub levels were also decreased in these mutants with values ranging from 16 to 29% of WT (*Figure 5A*, right panel; *Figure 5—figure supplement 1A*). As expected, higher levels of H2B-Ub are observed upon deletion of *UBP8*, encoding the H2B deubiquitinase. In contrast, upon activation of the Spt3/20 or Spt3/7 degrons, H3K18-Ac was >70% of WT levels after 30 min and thereafter decreased slowly over time (*Figure 5B*, left panel; *Figure 5—figure supplement 1B*). At 6 hr after IAA addition to these strains, the levels of H3K18-Ac mark in bulk chromatin were reduced to 45–50% of WT. The levels of H2B-Ub did not show any consistent changes during this time course (*Figure 5B*, right panel). The weak H2B-Ub antibody reactivity likely contributes to the variable H2B-Ub signals.

To investigate changes in histone acetylation at individual promoters, we next examined changes in chromatin-associated H3K18-Ac marks using ChIP-seq (*Figure 5C*, *Figure 5—figure supplement 2*) (*Supplementary file 7*). As expected, the bulk of this signal is associated with the +1 nucleosome (*Figure 5—figure supplement 2A*). We found a > 10 fold average defect in promoter-linked H3 acetylation in the *gcn5Δ* and *spt7Δ* strains and a 3.3-fold defect in the *spt3Δ* strain. In contrast, after 30 min of SAGA depletion, we observed only a 1.7-fold reduction. These changes in H3 acetylation were nearly the same at both the TFIID-dependent and coactivator-redundant genes (*Figure 5—figure supplement 2B*), consistent with earlier findings that SAGA contributes to chromatin modifications at all expressed genes (*Bonnet et al., 2014*). As expected, little or no H3 acetylation change was observed in the *ubp8Δ* strain. Comparison of different experiments revealed very high consistency between results of *gcn5Δ*, *spt7Δ* and Spt3/7-degron experiments, while *spt3Δ* experiment showed a lower but still significant correlation (*Figure 5—figure supplement 2C*). Our combined results strongly suggest that defects in SAGA-regulated chromatin modifications, particularly defects in Gcn5-dependent H3 acetylation, are a major contributor to the genome-wide decrease in transcription observed at nearly all genes in the *SPT* deletion strains. Since these chromatin modifications change relatively slowly after rapid SAGA depletion, we propose that transcription changes caused by rapid SAGA depletion are primarily due to disrupting one or more direct roles of SAGA in promoting transcription such as TBP-DNA loading.

## TFIID and SAGA map to both TFIID-dependent and coactivator-redundant genes

As discussed above, there has been disagreement in prior work as to the location and amount of TFIID binding to different gene classes. Early formaldehyde-based crosslinking studies showed that Tafs are generally depleted at the 'SAGA-dominated' genes compared with the 'TFIID-dominated genes', while MNase-based ChEC showed similar binding to both gene classes. We revisited this question using our revised gene lists and an improved ChEC-seq method that includes higher stringency criteria to map the genome-wide binding of these two coactivators. To minimize non-specific MNase cleavage and to avoid over digestion at authentic binding sites, we modified the original ChEC MNase cleavage conditions, using 10-fold lower calcium concentrations, and limited the MNase digestion time to 5 min. All experiments use spike-in DNA to normalize the control and experimental samples for quantitation. ChEC DNA cleavage patterns are compared with MNase controls, generated from cells where free MNase with a nuclear import signal is expressed from a promoter with greater or equal activity as the factor under study. Detailed methods and the criteria for peak calling are described in Materials and methods.

Using our updated ChEC-seq method to map Taf1, Taf7 and Taf13 binding (four biological replicates each), we identified 2938, 3681 and 3723 bound mRNA promoters, with extensive overlap between binding sites mapped with different Taf-MNase fusions (*Figure 6—figure supplement 1A*) (*Supplementary file 8*). Promoters were considered bound if a promoter-associated peak was called in at least 3 of 4 replicates. *Figure 6A* shows average plots of Taf-ChEC and control signals at the bound promoters and representative gene browser tracks are shown in *Figure 6—figure*

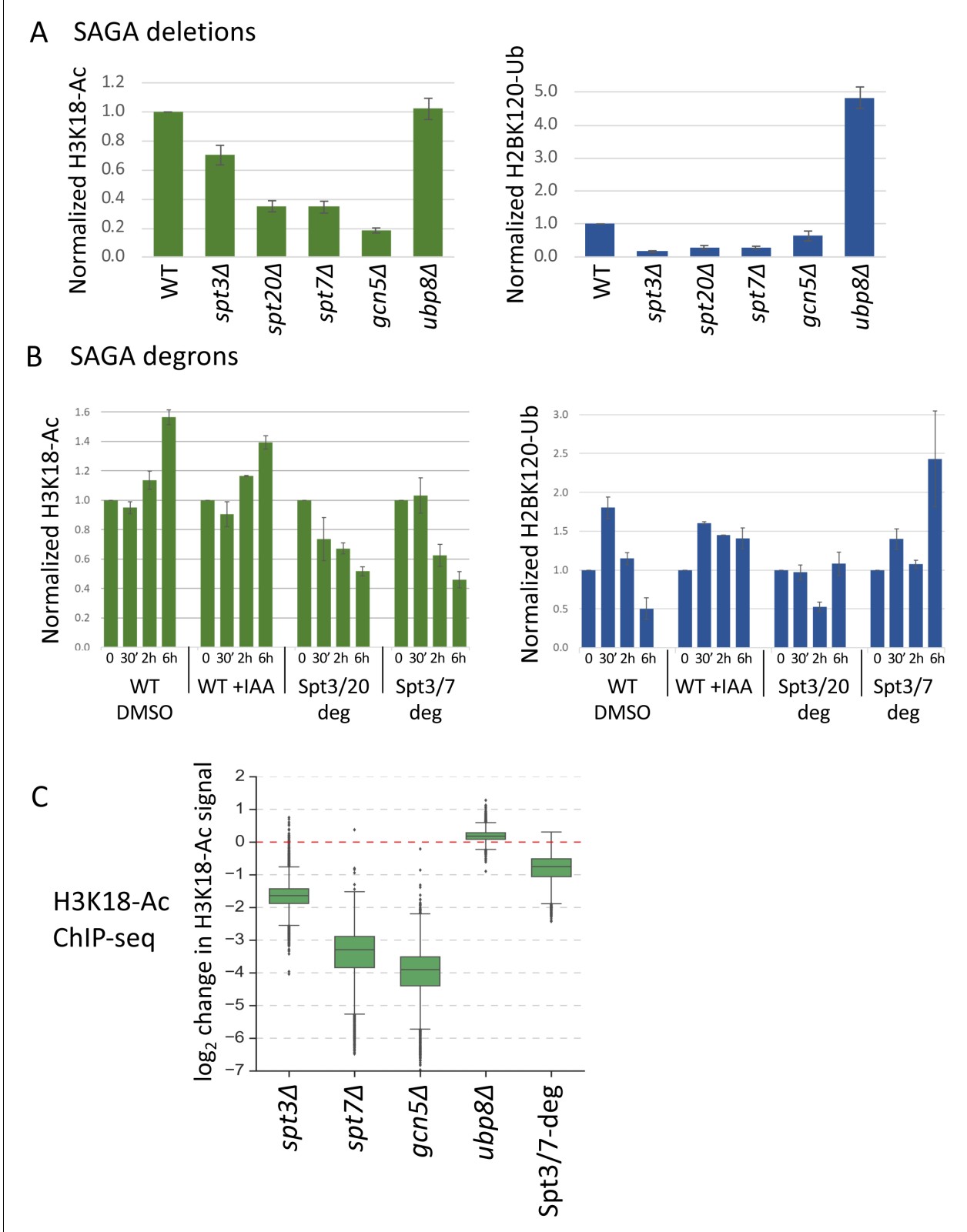

**Figure 5.** Histone modifications in *SPT* deletion and degron strains. (**A, B**) Quantitation of total H3K18-Ac and H2B-Ub in the *SPT* deletion and degron strains, normalized to the TFIIH subunit Tfg2 as a loading control. Data are from *Figure 5—figure supplement 1*. In panel (**B**), time after IAA or DMSO addition is indicated. (**C**) Box plot showing the ChIP-seq assayed log₂ change per gene in H3K18-Ac signal in the SAGA deletion mutants and Spt3/7 degron strain. Degron strain was treated for 30 min with IAA before formaldehyde crosslinking. Mean values from replicate experiments are plotted.
*Figure 5 continued on next page*

*Figure 5 continued*

The online version of this article includes the following source data and figure supplement(s) for figure 5:

**Source data 1.** Quantification of western blot results presented in *Figure 5—figure supplement 1*.
**Figure supplement 1.** Histone modifications in *SPT* deletion and degron strains.
**Figure supplement 2.** ChIP-seq analysis of H3K18-Ac signal in the SAGA deletion mutants and Spt3/7 degron strains.

*supplement 1B*). Using the above computational criteria,~75% of TFIID binding sites identified by ChIP-exo (*Vinayachandran et al., 2018*) overlap with the sites identified by Taf1-MNase, showing a close correspondence in results between our updated ChEC-seq approach and ChIP-exo (*Figure 6— figure supplement 1C*).

We next used ChEC-seq to revisit genome-wide binding of SAGA, where Spt3-MNase and Spt7-MNase identified 1601 and 3536 promoter binding sites (*Figure 6—figure supplement 1*) (*Supplementary file 8*). *Figure 6A* shows average plots of DNA cleavage from the Spt-MNase fusions and from free MNase. Our analysis showed that ~97% of SAGA binding sites mapped using Spt3-MNase overlap with the more extensive set of promoters mapped using Spt7-MNase (*Figure 6—figure supplement 1A*). Since Spt3 and Spt7 are thought to exist only in SAGA, the larger number of binding sites observed with Spt7-MNase likely indicates greater DNA accessibility of the MNase fusion.

Importantly, we found widespread promoter binding and $\geq$ 86% overlap in genes bound by TFIID and SAGA when comparing ChEC signals from Taf7 and Spt7-MNase fusions, supporting our findings above that most genes are regulated by both factors (*Figure 6B*). Further analysis showed little or no preference for the binding of TFIID or SAGA to either promoter class (*Figure 6C*). Taf13, Taf7, Taf1, and Spt7 MNase fusions all gave similar ratios of binding to each promoter type. The exception is the Spt3-MNase fusion that detected binding at fewer TFIID-dependent promoters compared with the other MNase fusions. However, since Spt3 is not known to be in any complex other than SAGA, we think it likely that this difference is due to different DNA accessibility and/or the architecture of SAGA and other factors at the two promoter types. Taken together, the widespread mapping of TFIID and SAGA at both promoter types is consistent with SAGA and TFIID acting at nearly all genes.

## Rapid SAGA depletion primarily affects maintenance of ongoing transcription rather than gene activation by Gcn4

Prior work showed that SAGA subunit deletion strains are defective in transcription activation at some genes (*Bhaumik and Green, 2001*; *Dudley et al., 1999*; *Mohibullah and Hahn, 2008*). To extend this work, we asked whether rapid SAGA depletion led to defects in activation of transcription (defined here as an increase in transcription in response to a stress signal) and/or in the maintenance of already ongoing transcription. Strains were grown in synthetic glucose media and RNA labeled for 5 min with 4-thioU followed by RT qPCR analysis of newly-synthesized mRNA. Transcription was first measured in either the *spt3Δ* or *spt7Δ* strains before or after addition of sulfometuron methyl (SM), which induces amino acid starvation and induction of Gcn4-dependent genes (*Figure 7A*). Transcription was measured at three TFIID-dependent genes (*ACT1, RPS5, SSH1*) that are not Gcn4 targets and three coactivator-redundant genes that are direct Gcn4 targets (*ARG3, HIS4, ARG5*). Compared with a wild type strain, the *SPT* deletion strains had ~ 3 fold lower expression levels at all three TFIID-dependent genes (compare light blue bars to light gray and light orange bars in *Figure 7A*). As expected, In the *SPT* deletion strains, these genes showed little or no transcription increase upon SM addition. At the three Gcn4 target genes, uninduced transcription was decreased > 4 fold compared with WT and the response to SM was dependent on the *SPT* deletion. In the WT strain, activation of *ARG3, HIS4*, and *ARG5* ranges from 2.6 to 3.8-fold (*Figure 7A*, compare light and dark blue bars). Although the levels of both uninduced and activated transcription were reduced in the *spt7Δ* strain, the three Gcn4-dependent genes showed some activation (1.7 to 2.5-fold; light vs dark gray bars), while activation was eliminated in the *spt3Δ* strain (light vs dark orange bars).

To determine the consequence of rapid SAGA depletion before transcription was induced, cells containing the Spt3/7 double degrons were first treated with IAA for 30 min to deplete these

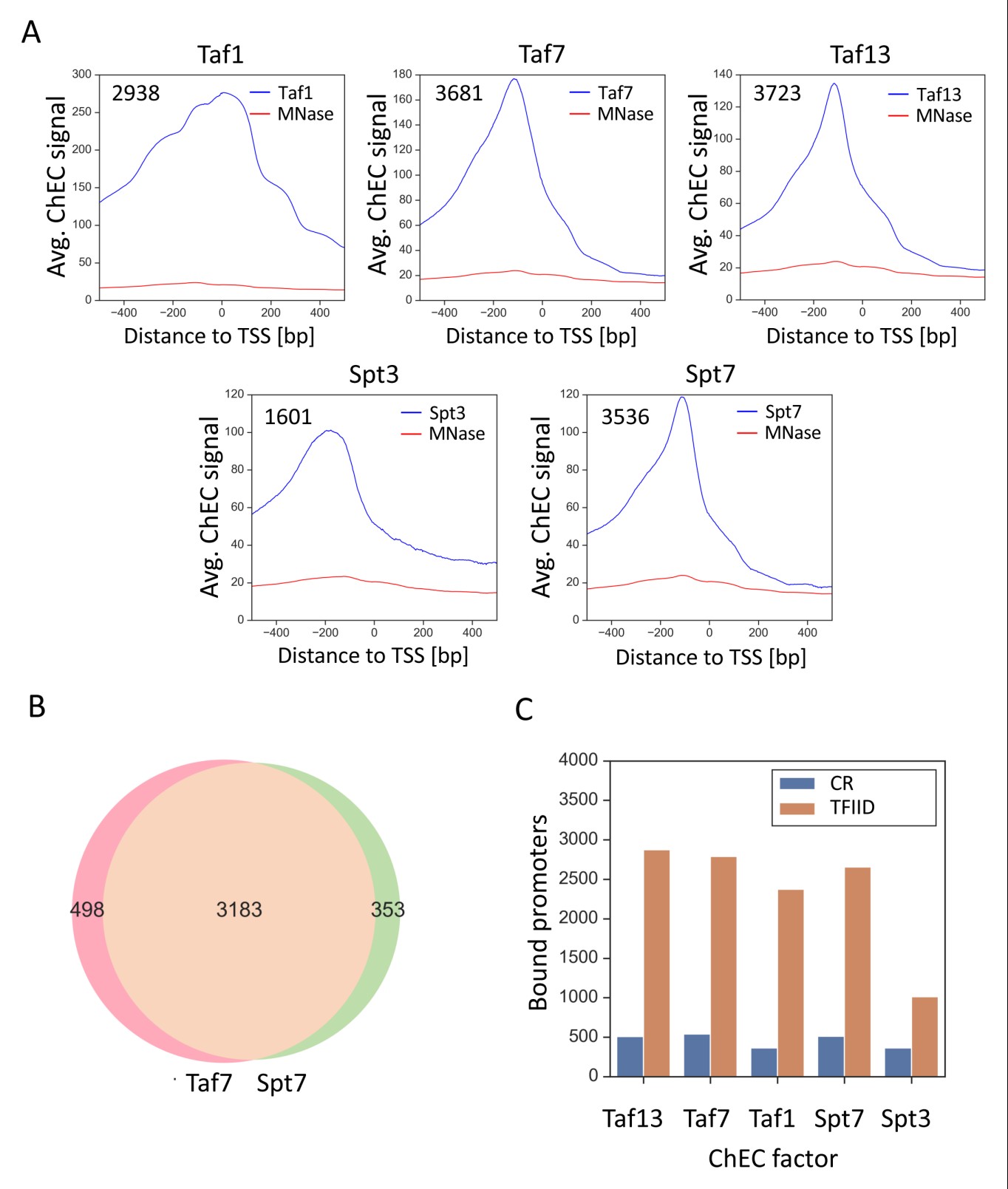

**Figure 6.** TFIID and SAGA show widespread binding to both gene classes. (**A**) Average plots of Taf1, Taf7, Taf13, Spt3 and Spt7 ChEC DNA cleavage versus free MNase cleavage. Signals were averaged for bound promoters only. (**B**) Venn diagram showing the overlap of bound promoters in Taf7 and Spt7 experiments. (**C**) Number of promoters bound by the indicated subunit of TFIID or SAGA with genes separated into coactivator-redundant and TFIID-dependent gene categories.

*Figure 6 continued on next page*

*Figure 6 continued*

The online version of this article includes the following figure supplement(s) for figure 6:

**Figure supplement 1.** Different TFIID and SAGA subunits show extensive overlap of binding sites.

proteins and then induced with SM for 60 min. This treatment still allowed for greater than normal fold-activation upon subsequent SM addition (3.6 to 6.9-fold; *Figure 7B*; light vs dark orange bars; *Figure 7—figure supplement 1*). In contrast, IAA added to already SM-induced cells reduced Gcn4-dependent transcription between 1.7 and 2.6-fold (light vs dark gray bars). Combined, our results show that, for Gcn4-dependent gene activation, the gene-specific role of SAGA is most important for efficiency or maintenance of ongoing transcription rather than for induction of transcription in response to an activation signal.

## Discussion

Here we have investigated the genome-wide and gene-specific roles for the coactivators SAGA and TFIID. Several prior studies suggested that, while TFIID and SAGA make genome-wide contributions, transcription at nearly all genes is dominated by one or the other factor and that many highly expressed genes show only modest TFIID-dependence (*Huisinga and Pugh, 2004*; *Petrenko et al., 2019*). This view seemingly conflicts with other studies where TFIID and SAGA were found to have major genome-wide roles at most genes (*Baptista et al., 2017*; *Bonnet et al., 2014*; *Warfield et al., 2017*). Our new results bring together many of these findings and provide a new mechanistic explanation for many genes that showed significant Taf-independent expression in earlier studies: TFIID and SAGA function is substantially redundant at these genes, with WT expression levels promoted by both coactivators.

The expression of 4900 genes were analyzed for coactivator dependence (83% of all yeast protein coding genes that met our expression and reproducibility threshold). Rapid subunit depletion revealed two classes of genes with different dependencies on TFIID and SAGA. The first class, termed TFIID-dependent genes, contains 87% of the analyzed genes and shows strong transcription dependence on TFIID but little or no change in response to rapid depletion of SAGA. The second class, termed coactivator-redundant genes, contains 13% of analyzed genes and shows a modest transcription decrease in response to rapid depletion of either TFIID or SAGA. Importantly, simultaneous depletion of both TFIID and SAGA leads to a severe transcription defect at these coactivator-redundant genes. This latter result is consistent with prior general conclusions that the combination of a SAGA subunit deletion and TFIID ts mutation can result in greater transcription defects compared with either single mutation (*Huisinga and Pugh, 2004*; *Lee et al., 2000*). Together, our results demonstrate that TFIID and SAGA have partially redundant function at the coactivator-redundant genes and, as described more fully below, we envision that these genes can use either the TFIID or SAGA pathways to promote TBP binding (*Figure 8*).

We also found that expression of nearly all genes is dependent on another SAGA function that was revealed in cells that had grown many generations without SAGA. Our results link this genome-wide role to SAGA-regulated chromatin modifications - marks that are slow to change upon rapid SAGA subunit depletion. Deletions in either *GCN5* or *UBP8,* the SAGA subunits responsible for chromatin modifications, show genome-wide transcription defects that are much stronger than those caused upon rapid SAGA depletion.

Western analysis of total chromatin showed that the level of H3K18-Ac and H2B-Ub are reduced in the *SPT* deletion strains and change slowly upon SAGA subunit depletion. Importantly, ChIP-seq analysis found that the levels of promoter-localized H3K18-Ac was much lower in the *SPT* deletion strains compared to strains depleted of SAGA function for 30 min. Indirect effects, such as decreased expression of general regulatory factors and components of the basal Pol II machinery, etc., likely also contribute to lower genome-wide expression in the deletion strains. The two roles of SAGA seem additive, which readily explains why the coactivator-redundant genes are more sensitive to *SPT* deletions compared to the TFIID-dependent genes. Our findings show that the coactivator-redundant genes require both SAGA functions while the TFIID-dependent genes are primarily dependent only on the SAGA-mediated chromatin modifications.

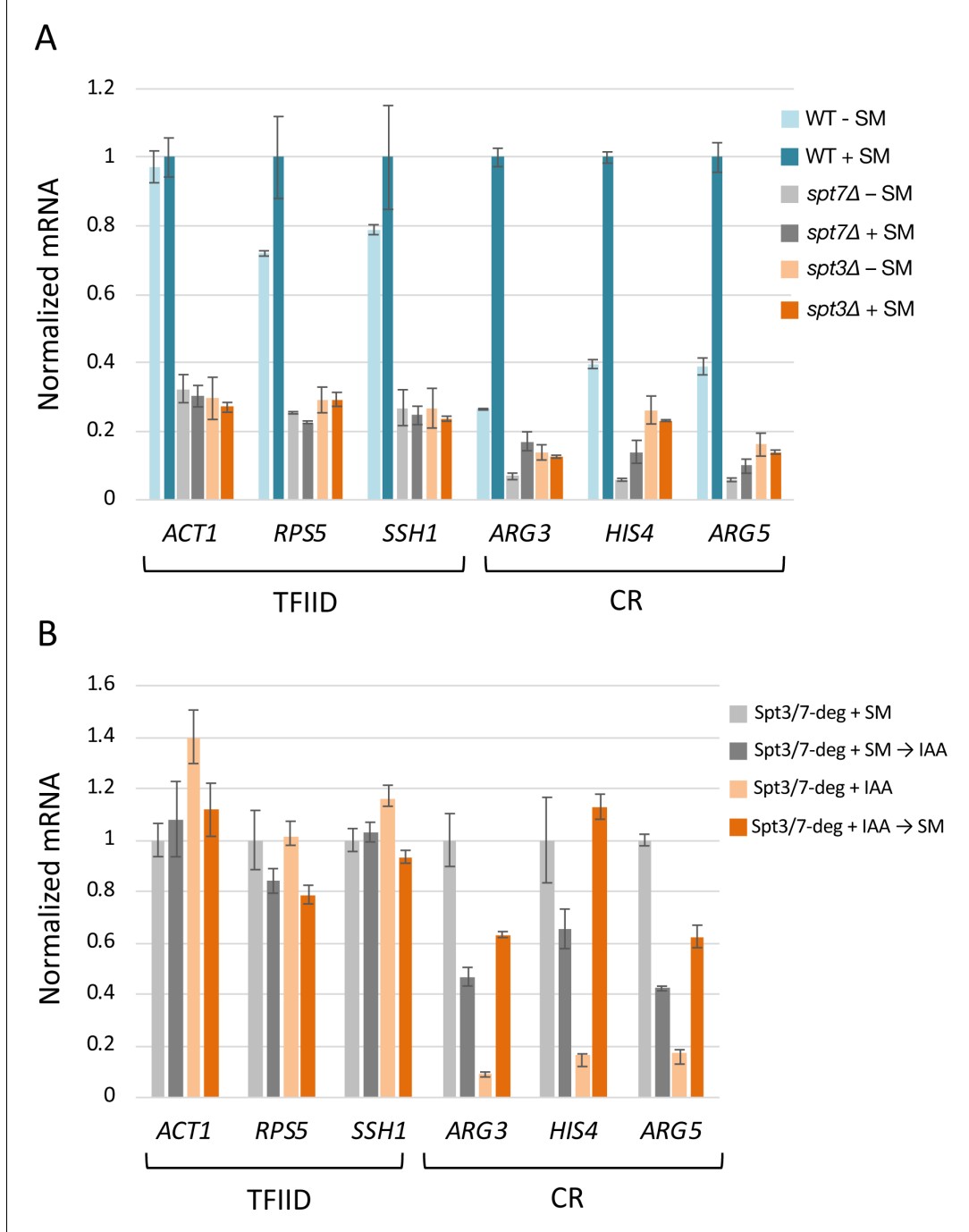

**Figure 7.** Rapid SAGA depletion does not prevent transcription activation by Gcn4. (**A**) RT-qPCR analysis of 4-thio Uracil labeled RNA purified from indicated *SPT* deletion strains in the presence or absence of SM or (**B**) from *SPT*-degron strains induced with SM either before or after treatment with IAA. Samples were normalized by spike-in of labeled *S. pombe* cells before RNA isolation, and RT qPCR values from each *S. cerevisiae* gene were normalized to the value of the *S. pombe* tubulin gene. Each experiment was performed in two biological and three technical replicates. Error bars represent the standard error of the mean for two biological replicates (values obtained after averaging results for three technical replicates).
The online version of this article includes the following source data and figure supplement(s) for figure 7:

**Source data 1.** Data from RT-qPCR analysis used for the plots in *Figure 7*.
**Figure supplement 1.** Degron efficiency in synthetic complete media with or without SM induction and IAA addition.

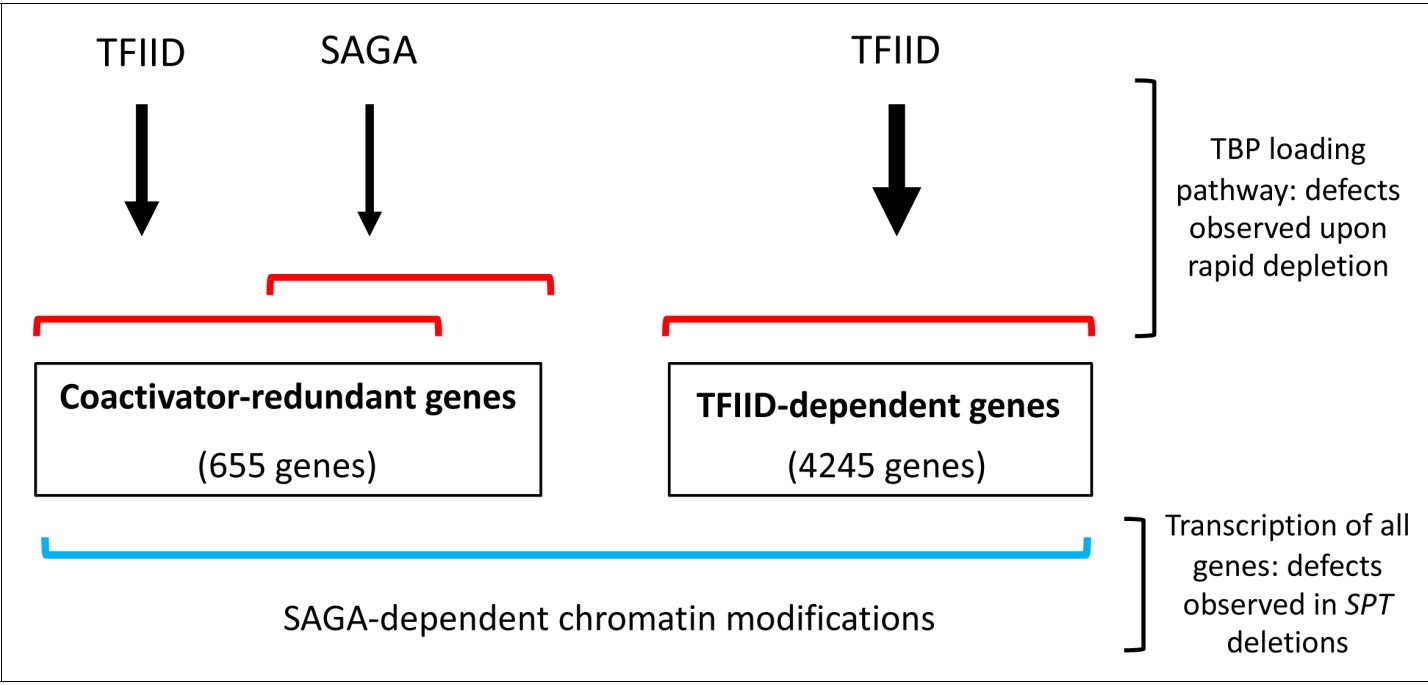

**Figure 8.** Model for roles of TFIID and SAGA and two different gene classes. Of the ~83% of yeast protein-coding genes analyzed in this study (4900 genes), the number of genes in each class is shown. Expression of genes in the TFIID set is strongly TFIID-dependent with little or no change after rapid SAGA depletion. TFIID and SAGA both contribute to expression of the coactivator-redundant genes where these factors have substantial overlapping function as revealed by rapid depletion experiments. Prior work suggests that these effects are due in part to the TBP-promoter DNA loading functions of TFIID and SAGA. Expression of nearly all genes is dependent on a separate function of SAGA, revealed by deletions in SAGA subunit genes, that is strongly linked to chromatin modifications that change slowly after rapid SAGA depletion.

Yeast cellular transcription is dominated by a small number of exceptionally highly expressed genes and it was proposed that highly expressed genes are biased for Taf-independent expression (*Petrenko et al., 2019*). We do find that the coactivator-redundant genes, while spread throughout the range of expressed genes, are enriched in the top 10% of transcribed genes. However, analysis of the ratio of SAGA-dependence/Taf-dependence shows that, within this entire gene class, there is no obvious relationship between coactivator sensitivity and gene expression. In fact, this analysis shows that most genes in the coactivator-redundant gene class show more reliance on TFIID compared with SAGA.

Results from mapping the genome-wide locations of TFIID using an improved and more stringent ChEC-seq method are consistent with our findings that TFIID functions at both the TFIID-dependent and coactivator-redundant genes. We observe no bias in promoter type bound by either TFIID or SAGA. In contrast, prior studies have observed a bias in Taf-promoter binding using formaldehyde crosslinking. At this time, the reason for this discrepancy is unknown, although we previously speculated that the observed bias in proximity of TFIID to nucleosomes at TFIID-dependent genes (*Rhee and Pugh, 2012*) may play a role in Taf-DNA crosslinking efficiency (*Warfield et al., 2017*).

To examine features leading to the different coactivator responses, we conducted a de novo motif search of promoters in each class. As expected, the TATA consensus sequence is enriched in the coactivator-redundant class, although nearly equal numbers of TATA-containing genes are found in the TFIID-dependent gene set. The results of de novo search found 14 transcription factor motifs that are enriched in one or the other gene class. However, this and more directed searches for these consensus motifs showed that no transcription factor binding site is exclusive for one or the other gene class. For example, while the motif for the Msn2/Msn4 factors is enriched in the CR gene class compared with all other genes, the majority of Msn2/Msn4 binding motifs are found in TFIID-dependent genes. From this combined analysis, we conclude that the distinction between the gene classes is due to multiple components rather than any one factor or promoter sequence. These factors likely

include the specific regulatory transcription factors used at each promoter, promoter sequence and perhaps chromatin architecture.

From the above results, it is clear that the coactivator-redundant genes are dependent on both coactivators rather than their expression being dominated by SAGA as proposed earlier; when one coactivator is eliminated, the other can at least partially compensate. What is the mechanistic basis for this behavior? Both factors were shown to have TBP-DNA loading activity. Based on earlier studies, we propose that the TFIID-dependent genes exclusively require Tafs for promoter recognition and TBP-DNA loading while the coactivator-redundant genes can use either SAGA or TFIID. Since many CR genes contain TATA elements, we envision that SAGA can recruit TBP more directly, taking advantage of: (1) specific TBP-TATA interactions and (2) direct TBP interactions with other TBP-compatible DNA binding motifs. This model can explain the surprising viability of strains with TBP mutations that are defective in TATA-DNA binding (*Kamenova et al., 2014*). Some of these TBP mutants may be defective in the SAGA-mediated TBP loading pathway but still function in the TFIID-dependent pathway. We note that these two gene categories do not perfectly correlate with the presence or absence of TATA. It has been shown, however, that the TATA consensus sequence does not perfectly correlate with function and that sequences surrounding TATA are also important (*Donczew and Hahn, 2018*; *Sprouse et al., 2008*).

Finally, we found that rapid depletion of SAGA, primarily decreases ongoing and already activated transcription at several coactivator-redundant genes. In contrast, rapid depletion does not affect activation per se at three Gcn4-dependent genes. This result is surprising, since it was shown earlier that *SPT* deletion strains have defects in gene activation (*Bhaumik and Green, 2001*; *Dudley et al., 1999*). However, we now know that these strains are defective in both functions of SAGA, unlike in our rapid depletion experiments. Gene activation (defined here as an increase in transcription in response to a stress signal) requires a rapid increase in the recruitment of TBP and other PIC components. This activation process seems intact after rapid SAGA depletion, based on the ~2 fold increased ratios of induced/uninduced transcription of Gcn4-dependent genes after SAGA depletion. It is important to note that, even though transcription is still inducible by Gcn4 after SAGA depletion, the absolute levels of basal and activated transcription are lower in these strains. We propose that these lower levels of basal and activated transcription explain why TFIID (without SAGA) can promote strong gene induction, while higher absolute levels of activated transcription in WT cells requires both coactivators.

From the available data, it is not yet clear why the coactivator-redundant genes require both SAGA and TFIID for efficient transcription. With the exception of the top 50 expressed genes, there are many genes in the top expression quintile that are TFIID-dependent and SAGA-independent. Therefore, the TFIID-directed pathway does not seem inherently limiting for relatively high transcription levels. Ongoing transcription, especially at very highly expressed genes, likely involves efficient transcription reinitiation via bursts of transcription. Efficient TBP-DNA loading may be important for this process at the coactivator-redundant genes and it will be of interest in future studies to probe these genes for defects in initiation, reinitiation and other steps in the transcription pathway after SAGA depletion.

## Materials and methods

### Strain construction

*S. cerevisiae* strains (*Supplementary file 9*) were constructed using standard yeast methods. Proteins were chromosomally tagged by high efficiency yeast transformation and homologous recombination of PCR-amplified DNA. Plasmid pFA6a-3V5-IAA7-KanMX6 (*Chan et al., 2018*) was used as the template for generating the IAA7 degron tags. This C-terminal tag contains three copies of the V5 epitope tag followed by the IAA7 degron (32 kDa total). For ChEC-seq experiments proteins were tagged with 3xFLAG-MNase::TRP1M × 6 using pGZ110 (*Zentner et al., 2015*). A strain expressing free MNase under control of the native *BDF1* promoter was constructed the following way. First, the *MED8* promoter in pSG79 (*Grünberg et al., 2016*) was exchanged with the *BDF1* promoter (containing 500 bp upstream DNA from the *BDF1* start codon). A XhoI/SacI fragment containing the $P_{BDF1}$-MNase fragment was inserted to the yeast integrating vector pRS303. The resulting pRD16 plasmid was linearized with BstEII and integrated into strain BY4705.

## Yeast cell growth

*S. cerevisiae* strains were grown as indicated in either rich media (YPD: 1% yeast extract, 2% peptone, 2% glucose, 20 µg/ml adenine sulfate) or synthetic complete (SC) media (per liter: 1.7 g yeast nitrogen base without ammonium sulfate or amino acids (BD Difco), 5 g ammonium sulfate, 40 µg/ml adenine sulfate, 0.6 g amino acid dropout mix (without -Ile -Val) and supplemented with two micrograms/ml uracil and 0.01% other amino acids to complement auxotrophic markers). Standard amino acid dropout mix contains 2 g each of Tyr, Ser, Val, Ile, Phe, Asp, Pro and 4 g each of Arg and Thr. *S. pombe* strains were grown in YE media (0.5% yeast extract, 3% glucose). Where indicated, *S. cerevisiae* strains at an A600 of ~ 1.0 were treated with 500 µM indole-3-acetic acid (IAA) dissolved in DMSO (or with DMSO alone) for 30 min. Where indicated, cells were incubated with 0.5 µg/ml sulfometuron methyl (SM) in DMSO (SM) or with DMSO alone for 60 min as described in the text and figure legends, prior to RNA labeling.

## Western blot analysis

1 ml cell culture was collected and pelleted from strains after treatment with IAA or DMSO, washed with 500 µl water, then resuspended in 75 µl yeast whole cell extract buffer. After heating for 5 min at 95°C, samples were centrifuged for 5 min at max speed, whole cell extracts were separated by SDS-PAGE and analyzed by Western blot using mouse monoclonal or rabbit polyclonal antibodies. To efficiently visualize histone marks, 5–10 ml of cell culture (adjusted to similar cell count based on OD measurement) was transferred to a tube on ice containing 2 ml 50 mM Tris, pH 7.5 + 10 mM NaN$_3$. Cells were pelleted, then resuspended in 75 µl SUTEB lysis buffer (10 mM Tris-HCl, pH 8.0, 1% SDS, 8M Urea, 10 mM EDTA, pH 8.0, 0.01% bromophenol blue) and transferred to a microcentrifuge tube, heated to 100°C for 4 min, then an equal volume of 0.5 mm zirconia/silica beads was added. Cells were lysed in a Mini Beadbeater-96 (BioSpec Products) for 30 s followed by 1 min rest on ice and repeated for a total of four cycles. 200 µl additional SUTEB lysis buffer was added, vortexed, and extracts were heated to 100°C for 1 min. Extracts were transferred to new tubes and stored at −70°C until separated by SDS-PAGE. Protein signals were visualized by using the Odyssey CLx scanner and quantified using Odyssey Image Studio software (Li-Cor) by generating a standard curve using a titration from WT extract. Each protein analyzed was normalized to the amount of the TFIIF subunit Tfg2. H2B-Ub was probed with antibody 5546 (Cell Signaling Technology) and H3K18-Ac with antibody 07–354 (EMD Millipore).

## RNA labeling and mRNA purification for RT-qPCR

Newly synthesized RNAs were labeled as previously described (*Bonnet et al., 2014*). 10 ml *S. cerevisiae* or 20 ml *S. pombe* cells were labeled with 5 mM 4-thiouracil (Sigma-Aldrich) for 5 min, the cells were pelleted at 3000 x g for 2 min, flash-frozen in liquid N$_2$, and then stored at −80°C until further use. *S. cerevisiae* and *S. pombe* cells were mixed in an 8:1 ratio and total RNA was extracted using the RiboPure yeast kit (Ambion, Life Technologies) using the following volumes: 480 µl lysis buffer, 48 µl 10% SDS, 480 µl phenol:CHCl$_3$:isoamyl alcohol (25:24:1) per *S. cerevisiae* pellet + 50 µl *S. pombe* (from a single *S. pombe* pellet resuspended in 850 µl lysis buffer). Cells were lysed using 1.25 ml zirconia/silica beads in a Mini Beadbeater-96 (BioSpec Products) for 3 min followed by 1 min rest on ice. This bead beading cycle was repeated twice for a total of 3 times. Lysates were spun for 5 min at 16K x g, then the following volumes combined in a 5 ml tube: 400 µl supernatant, 1400 µl binding buffer, 940 µl 100% ethanol. Samples were processed through the Ambion filter cartridges until all sample was loaded, then washed with 700 µl Wash Solution 1, and twice with 500 µl Wash Solution 2/3. After a final spin to remove residual ethanol, RNA was eluted with 25 µl 95°C preheated Elution Solution. The elution step was repeated, and eluates combined. RNA was then treated with DNaseI using 6 µl DNaseI buffer and 4 µl DNaseI for 30 min at 37°C, then treated with Inactivation Reagent for 5 min at RT. RNA was then biotinylated essentially as described (*Duffy et al., 2015*; *Duffy and Simon, 2016*) using 40 µl (~40 µg) total RNA and 4 µg MTSEA biotin-XX (Biotium) in the following reaction: 40 µl total 4-thioU-labeled RNA, 20 mM HEPES, 1 mM EDTA, 4 µg MTSEA biotin-XX (80 µl 50 µg/ml diluted stock) in a 400 µl final volume. Biotinylation reactions occurred for 30 min at RT with rotation and under foil. Unreacted MTS-biotin was removed by phenol/CHCl$_3$/isoamyl alcohol extraction. RNA was precipitated with isopropanol and resuspended in 100 µl nuclease-free H$_2$O. Biotinylated RNA was purified also as described (*Duffy and Simon, 2016*)

using 80 μl MyOne Streptavidin C1 Dynabeads (Invitrogen) + 100 μl biotinylated RNA for 15 min at RT with rotation and under foil. Prior to use, MyOne Streptavidin beads were washed in a single batch with 3 × 3 ml H₂O, 3 × 3 ml High Salt Wash Buffer (100 mM Tris, 7.4, 10 mM EDTA, 1 M NaCl, 0.05% Tween-20), blocked in 4 ml High Salt Wash Buffer containing 40 ng/μl glycogen for 1 hr at RT, then resuspended to the original volume in High Salt Wash Buffer. After incubation with biotinylated RNA, the beads were washed 3 × 0.8 ml High Salt Wash Buffer, then eluted into 25 μl streptavidin elution buffer (100 mM DTT, 20 mM HEPES 7.4, 1 mM EDTA, 100 mM NaCl, 0.05% Tween-20) at RT with shaking, then the elution step repeated and combined for a total of 50 μl. At this point, 10% input RNA (4 μl) was diluted into 50 μl streptavidin elution buffer and processed the same as the labeled RNA samples to determine the extent of recovery. 50 μl each input and purified RNA was adjusted to 100 μl with nuclease-free water and purified on RNeasy columns (Qiagen) using the modified protocol as described (*Duffy and Simon, 2016*). To each 100 μl sample, 350 μl RLT lysis buffer (supplied by the Qiagen kit and supplemented with 10 μl 1% βME per 1 ml RLT) and 250 μl 100% ethanol was added, mixed well, and applied to columns. Columns were washed with 500 μl RPE wash buffer (supplied by the Qiagen kit and supplemented 35 μl 1% βME per 500 μl RPE), followed by a final 5 min spin at max speed. RNAs were eluted into 14 μl nuclease-free water.

After purification of mRNA, one sample per batch of preps prepared in a single day was tested for enrichment of labeled RNA by RT qPCR, probing both unlabeled and labeled RNA from at least three transcribed genes. The purified 4TU RNA typically contained 2–10% contamination of unlabeled RNA. We also analyzed the RNA-seq data for enrichment of intron-containing RNA in the purified 4TU-labeled samples (*Figure 1—figure supplement 3B*). As expected, intron-containing RNA is clearly enriched in the newly synthesized RNA compared with total RNA.

## cDNA synthesis and quantitative PCR of newly-synthesized RNA

Two microliters RNA was used to generate cDNA using Transcriptor (Roche), random hexamer primer, and the manufacturer's instructions. cDNA was used either undiluted, 1/20, or 1/100 for quantitative PCR (qPCR) depending on the gene analyzed. Gene-specific qPCR was performed in triplicate using primers near the 5' end of the gene. qPCRs were assembled in 5 μl reaction mixtures in a 384-well plate using 2X Power SYBR green master mix (Thermo Fisher Scientific) and reactions were run on the QuantStudio5 Real-Time System (Thermo Fisher Scientific). Relative amounts of DNA were calculated using a standard curve generated from 10-fold serial dilutions of purified genomic DNA ranging from 10 ng to 0.001 ng. Relative amounts of *S. cerevisiae* transcript were normalized to *S. pombe* tubulin transcripts. All values are expressed relative to that of either wild type +SM or the Spt3/7 deg +SM strain, which were set to 1.0. Each experiment was performed in two biological replicates (cultures collected on separate days) and three technical replicates (independent measurements of the same qPCR sample). Values for technical replicates were averaged which gave a single value for each biological replicate. The final result is a mean of biological replicates.

## Preparation of 4thioU RNA libraries for NGS

Newly synthesized RNA isolated via 4-thioU labeling and purification was prepared for sequencing using the Ovation SoLo or Ovation Universal RNA-seq System kits (Tecan) according to the manufacturer's instructions and 1 ng (SoLo) or 50 ng (Universal) input RNA. Libraries were sequenced on the Illumina HiSeq2500 platform using 25 bp paired-ends at the Fred Hutchinson Genomics Shared Resources facility.

## ChEC-seq experiments

ChEC-seq was performed as previously described (*Grünberg et al., 2016*; *Zentner et al., 2015*) with several modifications. The final calcium concentration in the reaction mixture was 0.2 mM (2 mM in the original protocol) and MNase digestion was done for 5 min for all collected samples. Stop buffer was supplemented with *D. melanogaster* MNase-digested DNA (1 ng/ml stock concentration) in the amount calculated based on *S. cerevisiae* culture A600 measurement (volume = A600 x 8 μl).

## ChIP-seq experiments

ChIP-seq was performed similarly as described (*Rodriguez et al., 2014*) with the following modifications. Zirconia/silica beads were used for bead beating. Chromatin was sonicated for three rounds of

15 min. *S. pombe* (strain 972 hr) chromatin used as a spike-in was prepared the same way as *S. cerevisiae* samples (including cross linking, bead beating and sonication) and supplied by the Tsukiyama lab (Fred Hutch). Each IP reaction contained 1 µg of *S. cerevisiae* chromatin, 10 ng of *S. pombe* chromatin and 20 µl of Protein G Dynabeads (Invitrogen, #10004D) conjugated with 4 µl of anti-H3K18-Ac antibody (EMD Milipore, #07–354) and the final volume was brought to 500 µl with FA buffer. 25 µl of the *S. cerevisiae/S. pombe* chromatin mix was transferred to a separate tube as an 'input' sample before addition of beads and combined with 25 µl of the stop buffer. Chromatin-antibody complexes were eluted with two 25 µl volumes of stop buffer. IP and input samples were processed together from this point. After Proteinase K digestion, DNA was purified by phenol extraction (two step extraction for input samples) followed by ethanol precipitation in the presence of 20 µg glycogen. Pelleted DNA was washed with 70% ethanol and resuspended in 15 µl 10 mM Tris-HCl (pH 7.5).

## Preparation of sequencing libraries for ChEC-seq and ChIP-seq samples

Sequencing libraries were prepared similarly as described (*Warfield et al., 2017*) with several modifications. 1/6 vol of the final ChEC DNA sample and 2 ng of ChIP samples were used as an input. Final adapter concentration during ligation was 6.5 nM. Following ligation, in case of ChEC-seq samples two-step cleanup was performed using 0.25X vol AMPure XP reagent in the first step and 1.1X vol in the second step. 18 cycles were used for library amplification for ChEC-seq samples and 15 cycles for ChIP-samples. All libraries were sequenced on the Illumina HiSeq2500 platform using 25 bp paired-ends at the Fred Hutchinson Cancer Research Center Genomics Shared Resources facility.

## Analysis of DNA sequencing data

The majority of the data analysis tasks except sequence alignment, read counting and peak calling (described below) were performed through interactive work in the Jupyter Notebook (https://jupyter.org) using Python programming language (https://www.python.org) and short Bash scripts. All figures were generated using Matplotlib and Seaborn libraries for Python; (https://matplotlib.org; https://seaborn.pydata.org). All code snippets and whole notebooks are available upon request.

Paired-end sequencing reads were aligned to *S. cerevisiae* reference genome (sacCer3) and *S. pombe* reference genome (ASM294v2.20) with Bowtie2 (*Langmead and Salzberg, 2012*) using optional arguments '-I 10 -X 700 –local –very-sensitive-local –no-unal –no-mixed –no-discordant'. Details of the analysis pipeline depending on the experimental technique used are described below.

## Analysis of RNA-Seq data

SAM files for *S. cerevisiae* data were used as an input for HTseq-count (*Anders et al., 2015*) with default settings. The GFF file with *S. cerevisiae* genomic features was downloaded from the Ensembl website (assembly R64-1-1). Signal per gene was normalized by the number of all *S. pombe* reads mapped for the sample and multiplied by 10000 (arbitrarily chosen number). Genes classified as dubious, pseudogenes or transposable elements were excluded leaving 5797 genes for the downstream analysis. As a next filtering step, we excluded all the genes that had no measurable signal in at least one out of 48 samples including SAGA and TFIID degron experiments and *SPT3*, *SPT7* and *SPT20* deletion experiments (samples for simultaneous depletion of SAGA and TFIID, *GCN5* and *UBP8* deletion experiments and WT control auxin experiment were not used for this filtering step). The remaining 5158 genes were sorted by the average expression level based on combined results of all WT and DMSO samples (22 samples total). For this analysis, signal per gene was further normalized by the gene length. Results from all relevant samples were averaged and genes were ranked (*Supplementary file 1*). We used this information to filter out genes with the lowest expression after comparing average coefficient of variation for each sample at different cutoffs for the number of highly expressed genes left in the analysis. The biggest relative decrease in observed average CV was visible after filtering out 5% (258) lowly expressed genes. Consequently, 95% of 5158 genes (i.e. 4900 genes) were used in the rest of this work. The results of biological replicate experiments for each sample were averaged (*Supplementary file 2*). All experiments were done in triplicate except YPD Taf-degron and wild-type BY4705 samples (labeled SHY772) that were done in duplicate. Replicate experiments showed low variation for the majority of genes which is visualized in *Figure 1—*

*figure supplement 1* using coefficient of variation. Data for simultaneous depletion of SAGA and TFIID, *GCN5* and *UBP8* deletion experiments and WT control auxin experiment were limited to the same set of 4900 highly expressed genes and the results of biological replicates (two or three) were averaged. Corresponding samples were compared to calculate $\log_2$ change in expression per gene (IAA to DMSO samples for degron experiments and deletion mutant to WT strain for SAGA deletion experiments – strain BY4705 [*Brachmann et al., 1998*] was used as a background for *SPT20*, *SPT7*, *GCN5* and *UBP8* deletion strains, SHY565 (BY4705 with Rpb3-3xFlag::KanMX) was used to construct the *SPT3* deletion strain) and degron strains are derivatives of SHY1037 (*Warfield et al., 2017*) (*Supplementary file 9*).

## Analysis of ChEC-seq data

SAM files for *S. cerevisiae* data were converted to tag directories with the HOMER (http://homer.ucsd.edu, (*Heinz et al., 2010*) 'makeTagDirectory' tool. Peaks were called using HOMER 'findPeaks' tool with optional arguments set to '-o auto -C 0 L 2 F 2', with the free MNase data set used as a control. These settings use default false discovery rate (0.1%) and require peaks to be enriched 2-fold over the control and 2-fold over the local background. Resulting peak files were converted to BED files using 'pos2bed.pl' program. For each peak, the peak summit was calculated as a mid-range between peak borders. For peak assignment to promoters the list of all annotated ORF sequences (excluding sequences classified as 'dubious' or 'pseudogene') was downloaded from the SGD website (https://www.yeastgenome.org). The data for 5888 genes were merged with TSS positions obtained from *Park et al. (2014)*. If the TSS annotation was missing, the TSS was manually assigned at position −100 bp relative to the start codon. Peaks were assigned to promoters if their peak summit location was in the range from −300 to +100 bp relative to the TSS. In a few cases, where more than one peak was assigned to the particular promoter, the one closer to the TSS was used. The lists of bound promoters for replicate experiments were compared and only promoters bound in three out of four replicates were used in downstream analysis. In cases a promoter did not have a peak assigned in one of the replicates we used the position of the strongest signal around the TSS as the peak summit. Manual inspection of many of these cases confirmed the validity of this approach.

Coverage at each base pair of the *S. cerevisiae* genome was calculated as the number of reads that mapped at that position divided by the number of all *D. melanogaster* reads mapped in the sample and multiplied by 10000 (arbitrarily chosen number). Signal per promoter was calculated for each replicate experiment as a sum of normalized reads per base in a 300 bp window around the peak summit. The overall signal per promoter (*Supplementary file 8*) is the average signal from all replicate experiments.

FASTQ files for the Taf1 ChIP-exo were obtained from the SRA (SRR5511893) (*Vinayachandran et al., 2018*). The data were processed as described above except for the use of RPM normalization.

## Analysis of ChIP-seq data

Coverage at each base pair of the *S. cerevisiae* genome was calculated as the number of reads that mapped at that position divided by the number of all *S. pombe* reads mapped in the sample, multiplied by the ratio of *S. pombe* to *S. cerevisiae* reads in the corresponding input sample and multiplied by 10000 (arbitrarily chosen number). The list of all annotated ORF sequences (excluding sequences classified as 'dubious' or 'pseudogene') was downloaded from the SGD website (https://www.yeastgenome.org). The data for 5888 genes were merged with TSS positions obtained from *Park et al. (2014)*. If the TSS annotation was missing the TSS was manually assigned at position −100 bp relative to the start codon. H3K18-Ac signal per gene was calculated for each replicate experiment as a sum of normalized reads per base in a window between 200 bp upstream and 300 bp downstream from TSS. We excluded three genes which had no measurable signal in at least one sample. The results of biological duplicate experiments for each sample were averaged and the change in H3K18-Ac signal per gene was calculated by comparing corresponding samples – IAA to DMSO samples for degron experiments and deletion mutant to WT strain for SAGA deletion experiments. Strain BY4705 (*Brachmann et al., 1998*) was used as a background for *SPT7*, *GCN5* and *UBP8* deletion strains, SHY565 (BY4705 with Rpb3-3xFlag::KanMX) was used to construct the *SPT3*

deletion strain. The final dataset contains $\log_2$ change in H3K18-Ac signal values per gene for 5885 genes (*Supplementary file 7*).

## K-means clustering analysis

K-means clustering was performed using 'KMeans' function from Python sklearn.cluster library with default settings. Two clusters were found to give the best separation using silhouette analysis ('silhouette-score' function from sklearn.cluster library). $\log_2$ change in transcription values from all SAGA deletion experiments and all TFIID degron experiments were used and all 4900 genes which gave reproducible results in RNA-seq (as described above) were included in the analysis.

## De novo motif discovery

De novo motif discovery was performed using HOMER (http://homer.ucsd.edu) (*Heinz et al., 2010*) 'findMotifs.pl' program. Motif length was set to be in a range 6–16 bp and the promoter region was defined as the area between 400 bp upstream to 100 bp downstream from TSS. Promoters from the analyzed class were screened against all other yeast promoters.

Resulting motifs were screened against the library of known motifs from HOMER database and the best matches were reported (*Supplementary file 5*). Two most highly represented motifs for CR genes (TATA box and Msn2/4 binding site) were chosen for more detailed analysis. Each of these two motifs was searched for among 4900 promoters classified in this study. For TATA box a consensus TATAWAW was used and the search was limited to the region from 200 bp upstream to TSS. For Msn2/4 a consensus (A/C/G)AGGGG was used (*Stewart-Ornstein et al., 2013*) and the search was limited to the region from 300 bp upstream to 50 bp upstream relative to TSS. All promoters carrying at least one consensus sequence in a defined range were classified as either TATA-containing or Msn2/4-containing.

## Acknowledgements

We thank Gabe Zentner and Steve Henikoff for discussions, Christine Cucinotta and Brian Strahl for advice on SAGA-regulated chromatin modifications and detection, Wei Sun for discussion regarding clustering approaches, Matthew Fitzgibbon for helpful comments about RNA-seq data analysis, Toshi Tsukiyama for *S. pombe* chromatin, Steve Henikoff for *D. melanogaster* chromatin and Laszlo Tora, Didier Devys, and Toshi Tsukiyama for comments on the manuscript. Supported by NIH grants GM053451 and GM075114 to SH and NIH P30 CA015704 to the Fred Hutch Genomics and Computational Shared Resources facility.

## Additional information

### Funding

| Funder | Grant reference number | Author |
| --- | --- | --- |
| National Institutes of Health | RO1 GM053451 | Steven Hahn |
| National Institutes of Health | RO1 GM075114 | Steven Hahn |

The funders had no role in study design, data collection and interpretation, or the decision to submit the work for publication.

### Author contributions

Rafal Donczew, Conceptualization, Formal analysis, Validation, Investigation, Visualization, Writing - original draft, Writing - review and editing; Linda Warfield, Investigation, Visualization, Writing - original draft, Writing - review and editing; Derek Pacheco, Investigation, Writing - review and editing; Ariel Erijman, Formal analysis, Investigation, Writing - review and editing; Steven Hahn, Conceptualization, Supervision, Funding acquisition, Investigation, Writing - original draft, Project administration, Writing - review and editing

Author ORCIDs

Rafal Donczew (iD) https://orcid.org/0000-0001-9729-4153

Steven Hahn (iD) https://orcid.org/0000-0001-7240-2533

Decision letter and Author response

Decision letter https://doi.org/10.7554/eLife.50109.sa1

Author response https://doi.org/10.7554/eLife.50109.sa2

## Additional files

### Supplementary files

- Supplementary file 1. Spike-in normalized signal for all genes (5158) which had detectable transcription in 48 RNA-seq samples collected in this study (samples for simultaneous depletion of SAGA and TFIID, *GCN5* and *UBP8* deletion experiments and WT control auxin experiment were not used for this analysis). Expression column is the average signal for all DMSO and WT samples. This value was further normalized by the gene length to give normalized expression which was used to sort the genes from the highest to the lowest expression.

- Supplementary file 2. Average spike-in normalized signal for replicate experiments for the final set of 4900 genes analyzed in this study and average expression per gene (based on DMSO and WT experiments). Rows are sorted by the average expression.

- Supplementary file 3. Average $\log_2$ changes in transcription in the degron and deletion strains, results of k-means clustering and average expression per gene (based on DMSO and WT experiments). Rows are sorted by the average expression.

- Supplementary file 4. Average $\log_2$ changes in transcription from degron experiments simultaneously depleting SAGA and TFIID components (Spt3/Taf13 and Spt7/Taf13). Data for other strains, results of k-means clustering and average expression per gene are the same as shown in *Supplementary file 3*. Rows are sorted by the average expression.

- Supplementary file 5. Motif enrichments found in the TFIID-dependent and coactivator-redundant gene classes.

- Supplementary file 6. Average $\log_2$ changes in transcription for *GCN5* and *UBP8* deletion experiments. Data for other strains, results of k-means clustering and average expression per gene are the same as shown in *Supplementary file 3*. Rows are sorted by the average expression.

- Supplementary file 7. Average $\log_2$ changes in H3K18-Ac signal in the SAGA deletion mutants and Spt3/7 degron strain.

- Supplementary file 8. Average spike-in normalized ChEC signals at bound promoters for the following MNase-fusions: Taf1, Taf7, Taf13, Spt3 and Spt7. Genes in each table are sorted by signal intensity.

- Supplementary file 9. *S. cerevisiae* and *S. pombe* strains used in this study. Strains were validated using a combination of genetic assays, phenotypic analysis, Western analysis, PCR analysis and nucleic acid sequencing.

- Transparent reporting form

### Data availability

The data discussed in this publication have been deposited in NCBI's Gene Expression Omnibus and are accessible through GEO Series accession number GSE142122.

The following dataset was generated:

| Author(s) | Year | Dataset title | Dataset URL | Database and Identifier |
| --- | --- | --- | --- | --- |
| Donczew R, Warfield L, Erijman A, Pacheco D, Hahn S | 2020 | Two separate roles for the transcription coactivator SAGA and a set of genes redundantly regulated by TFIID and SAGA | https://www.ncbi.nlm.nih.gov/geo/query/acc.cgi?&acc=GSE142122 | NCBI Gene Expression Omnibus, GSE142122 |

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
