## [Decision Letter]

**Acceptance summary:**

This work elegantly revisits the highly-debated topic of coactivator dependency of different yeast promoter classes. Using acute depletion of TFIID and SAGA subunits and nascent transcript labeling, it reveals that there is a large class of genes that is TFIID dependent and a smaller TFIID/SAGA-redundant class. Importantly, this work clearly shows that acute depletion has different effects on gene expression compared to deletion of the subunits, likely due to the fact that SAGA dependent modifications remain after acute depletion of SAGA, addressing some of the previous confusion in the field as to coactivator dependency of gene classes. The conclusions advance our understanding of the relative roles of SAGA and TFIID in gene regulation in yeast. The work will be of general interest to the transcription/chromatin community and will hopefully lead to future studies identifying the mechanistic basis of the different coactivator dependencies for different gene classes.

**Decision letter after peer review:**

[Editors’ note: the authors submitted for reconsideration following the decision after peer review. What follows is the decision letter after the first round of review.]

Thank you for choosing to send your work, "Two separate roles for the transcription coactivator SAGA and a set of genes redundantly regulated by TFIID and SAGA", for consideration at *eLife*. Your initial submission has been assessed by a Senior Editor in consultation with a member of the Board of Reviewing Editors. Although the work is of interest, we regret to inform you that the findings are not appropriate for further consideration at *eLife*.

Specifically, based on all the reviews, the paper has some experimental, formatting, and writing issues that are addressable in a reasonable time frame. Importantly, some of these issues, when addressed, could change the conclusions made. For example, if histone modification levels change on the SAGA/TFIID-redundant genes when measured by ChIP, the conclusions on how SAGA depletion affects transcription would need to be modified. Certainly, the concerns about levels of depletion and growth defects of the various mutants needs to be addressed. However, even when all the experimental and writing concerns are corrected, the question remains about whether the paper makes a large enough contribution for our journal. We felt that this topic has been visited and revisited numerous times, and we are not completely certain this paper is giving us the final answer. The impact would be higher if you had included ChEC-seq data, although reading of the accompanying paper indicates that the conclusions about SAGA occupancy from the Molecular Cell paper from your lab would be upheld. We feel that the big remaining question is what mechanistically distinguishes the two gene classes? You have attempted to address this but, in our opinion, the mechanistic distinctions remain unclear. Even though this paper does a better job than many characterizing which promoters go in which category, we still didn't come away with any new insights into why some respond to SAGA and most don't.

We do not intend any criticism of the quality of the data or the rigor of the science. Indeed, this is clearly very nice quality work and should be published, but in a more specialized journal, in our opinion.

Reviewer #1:

This manuscript is the next installment of a series of papers over several years that have attempted to discern which yeast genes are regulated by the two coactivators SAGA and TFIID. As different techniques have been used, the view has changed. In these current experiments, the authors use metabolic labeling with 4-thioU to measure the level of transcription under either mutant conditions or after depletion of SAGA or TFIID components. Interestingly, they obtain very different results for SAGA between the depletion and deletion conditions. From the depletion studies, they conclude that approximately 13% of the yeast genes that they could assay are regulated in a partially redundant fashion by both SAGA and TFIID, while the remainder are primarily regulated by TFIID. The interpretation is reasonable, although their deletion studies show a much greater effect of SAGA mutants and their ChEC studies, in agreement with previous studies, show that SAGA is present at most promoters, as is TFIID. Overall, this is nicely done work that should be of interest to those in the transcription field. Several comments on both the presentation and the experiments are below.

1) The first sentence of the Abstract is inaccurate as SAGA does more than chromatin modifications, as the authors reveal in the Introduction. I suggest revising the sentence to the following: "Deletions of genes encoding subunits of the yeast coactivator SAGA reveal a general role in transcription linked to the multiple activities of this complex."

2) Please edit throughout the manuscript to correct little writing errors. As one example, in the Abstract, "…87% of expressed genes we term "TFIID-dependent" are highly sensitive to TFIID depletion…" the word "that" should be inserted before "we term." Also, change "contacts" to "contact." Here's another example: – "The genome-wide specificity of SAGA and TFIID have been…" – either change specificity to specificities or change have to has. There are more cases like this throughout the manuscript.

3) Introduction section – "relatively weak TBP binding" might be biochemically accurate but it implies a weak activity with respect to the degree of gene regulation, which is not the case for spt3 and spt8 mutants.

4) Figure 1 and accompanying results – In the comparisons of deletions versus depletions, it would be helpful if the authors clarified their choices for comparison since they are comparing single gene deletions to multiple subunit depletions. Presumably they chose to deplete multiple SAGA subunits at once to increase the chances of efficient SAGA depletion, but please state that explicitly of that is the case. In an ideal world, they would have compared deletion/depletion of the same subunits, such as Spt3 vs Spt3 or Spt3+Spt7 vs Spt3+Spt7. It's formally possible, for example, that depletion of only Spt3 would have a more severe phenotype than the double depletion, although this is unlikely based on previous studies. If they have any unpublished evidence about this, they might mention it.

5) Figure 1—figure supplement 4C – What is being correlated in this table? If each experiment was generally done in triplicate, for each genotype, what was used to general the correlation coefficient? Please explain in the legend.

6) Since Tafs are essential for viability, what was done to ensure that the cells were still viable during the RNA measurements, for either Taf depletions or Taf + SAGA depletions?

7) Regarding depletion versus deletion, it seems possible that the difference is caused by the level of depletion rather than the time without the protein. Even though the authors have tried to rule this out, I think the formal possibility should be mentioned.

8) Subsection “SAGA-regulated chromatin modifications change little during rapid SAGA inactivation” – For SAGA's role in activation, the authors should also cite three other references – Larschan et al., 2001, Bhaumik et al., 2001, and Dudley et al., 1999.

9) Figure 4 and subsection “SAGA-regulated chromatin modifications change little during rapid SAGA inactivation” – The authors state that it was unexpected to observe lower H2B-Ub levels in some of the SAGA mutants in panel A. Similarly, it was unexpected to observe lower H3K18-Ac levels in the spt3 deletion strain, based on previous genetic, biochemical, and structural studies that strongly suggest that the HAT module is physically and biochemically distinct from Spt3. Therefore, the same reasoning used for the effect on H2B-UB might apply for the reduction in H3K18-Ac levels in the spt3 mutant. If the authors wish to impose histone modifications as a reason for the depletion vs. deletion difference (Discussion paragraph three), this might be mentioned, as an spt3 deletion has a severe effect on transcription yet in theory it should not affect the level of histone modifications.

10) Figure 5, subsection “Rapid SAGA depletion primarily affects maintenance of ongoing transcription rather than gene activation by Gcn4” paragraph two, – This was an interesting experiment. In contrast to what the authors write, the fold induction looks to be greater in the IAA-treated samples in 5B compared to the non-IAA treated cells in 5A, particularly for ARG3 and HIS4.

11) Figure 1 and 2 legends and probably elsewhere – SC media is stated as the growth medium but it is not described in the Materials and methods. Please clarify.

12) The authors need to provide a list of the yeast strains, including names and genotypes.

13) The authors discuss the SAGA-TFIID redundancy in the Discussion but they don't offer an explanation for two aspects of the redundancy. First, why is it partial? If either complex can work at those genes, why does expression decrease at all? Is it due to insufficient levels of the other complex? Second, what does the reduced expression represent? Is it caused by normal levels from a subpopulation of cells in the culture, or is it due to reduced expression from all of the cells? In future experiments, this could be addressed by single cell analysis.

Reviewer #2:

In this manuscript, Donczew et al. revisit the highly-debated topic of yeast promoter classes, using acute depletion of TFIID and SAGA subunits and nascent transcript labeling. Older studies by the Pugh lab categorized yeast promoters as SAGA-dominated and TFIID-dominated. In general, the SAGA-dominated genes were described as TATA-enriched and stress responsive, while the TFIID-dominated genes lacked consensus TATA boxes. In a 2017 Molecular Cell paper (Baptista et al.,), Hahn and collaborators showed that deletion of SAGA subunits significantly reduced nascent transcript levels at both TFIID- and SAGA-dominated gene classes and also showed that SAGA occupies the promoters of both gene classes by ChEC-seq. This paper threw into question the classification of promoters by coactivator dependencies and suggested that the functions of SAGA are more widespread than previously thought. However, recently, the ChEC-seq profiles reported in the Baptista paper have been disputed by the Pugh lab in a paper in press at Molecular Cell.

After acutely depleting SAGA and TFIID subunits and measuring nascent transcription, the authors in this paper conclude that there are two major classes of Pol II promoters with respect to TFIID and SAGA dependencies: a large TFIID-dependent class and a smaller TFIID/SAGA-redundant class, which accounts for 13% of promoters tested. The authors define these classes through measurements of nascent RNA changes in response to factor depletion using the auxin-inducible degron system, followed by k-means clustering. By co-depleting Taf13 and either Spt3 or Spt7, the authors provide strong support for the redundancy model. They also report results from a new ChEC-seq protocol which show widespread TFIID and SAGA occupancy at all genes in line with the Baptista paper. Regarding the mechanistic distinctions between the TFIID-dependent and TFIID/SAGA-redundant promoters, the authors conclude that gene expression levels do not dictate whether a promoter is redundantly regulated by the coactivators. By reanalyzing the TATA sequences in the promoter classes, the authors report that TFIID/SAGA-redundant promoters are not biased toward TATA-containing versus TATA-less. Western data show that two histone modifications regulated by SAGA do not greatly change during the AID depletion time course. From this the authors conclude that some other function of SAGA, such as a protein-protein interaction, is being disrupted by the depletion and leading changes in transcription. Using RT-PCR, the authors test the effect of SAGA depletion on activation of three Gcn4-regulated genes and find that activation is unaffected, whereas ongoing transcription is reduced.

The main conclusion from this paper is the redefining of yeast promoters using acute depletion and nascent transcript labeling. The results support a large TFIID-dependent class and smaller class that can utilize either TFIID or SAGA for their regulation. The molecular basis of the two classes defined here remains uncertain.

Specific comments:

1) The identification of two promoter classes is the main conclusion of the paper and is based on acute protein depletion using the AID system, 4-thioU RNA-seq, and k-means clustering. The paper requires additional controls to show that the AID tags are not disrupting protein function, that auxin or DMSO isn't affecting expression of an untagged WT strain, and that nascent transcripts are enriched (such as by showing browser tracks of intron transcripts). Of note, in Figure 4B, levels of H2B-Ub at the 0 min time point are lower than the WT control. Are the tags affecting the deubiquitination function of SAGA? Details of the clustering analysis are lacking, and a heatmap showing the clusters would be useful. Details on what the authors call an effect on transcription are lacking. Was every gene that passed the filter described in the Materials and methods included in the clustering regardless of the magnitude of the effect on transcription?

2) The colors highlighting the two classes of promoters are switched in the middle of the paper. This needs to be corrected.

3) Figures 2, 3 and Figure 3—figure supplement 1: The axis label "gene expression rank" is confusing. I assume the #1 gene is that with the highest expression but this is not clearly stated.

4) Figure 3A. The authors conclude that the classification of TFIID/SAGA-redundant genes is not based on gene expression levels. This seems to be an overstatement. If the authors included percentages in panel A, it would seem clear that the TFIID/SAGA-redundant genes represent a larger percentage of the top quintile than the lower quintiles.

5) Figure 3D. It is unclear how robust the distinction between TATA+ and TATA-less promoters is. Now that the authors have discrete gene sets, why not perform a de novo motif analysis to identify TATA or TATA-like elements as well as other cis-regulatory information that might underlie the coactivator requirements? This type of analysis could provide some mechanistic understanding of the gene classes, which is currently lacking from the field.

6) Figure 4. How many repetitions were performed and used to calculate the western signals? In panel B, were the lanes on the left spliced from the same blot, exposed for the same length of time? Ubiquitination is unstable and typically extracts are prepared in denaturing buffers. Information on extract preparation is needed.

7) Figure 4. From the western analysis, the authors conclude that loss of SAGA-dependent histone modifications cannot explain the changes in transcription that occur in response to SAGA depletion. However, because only 13% of promoters are dependent on SAGA, it is unclear if changes in SAGA-dependent histone modifications would be visible by western. ChIP experiments are needed to test the histone modifications at promoters in each class upon factor depletion.

8) Figure 5. More experimental details are needed.

9) Figure 6. Except for changes in color, both of these panels are the same as those included in the Donczew et al. paper on modified ChEC-seq that the authors cite as submitted for publication and was provided to the reviewers of this paper. This is clearly unacceptable. Frankly, I feel that the authors should include the modified ChEC-seq protocol, analysis, and results in this paper and not in a separate paper.

10) Figure 1—figure supplement 4C. Correlations between data sets are more standardly shown as xy plots. This graph is confusing. What does "genes left" mean?

11) Figure 3—figure supplement 1A. A log scale on the y-axis would be better.

Reviewer #3:

It's hard to believe we're still arguing about how many and which kinds of genes use TFIID and/or SAGA. This debate goes back to the mid-1990s, when Struhl and Green used TS alleles to question whether TAFs really had a role in activation. In 2004, the Pugh lab classified about 10% of genes as "SAGA-dominant" and the rest as "TFIID-dominant". The arguments continue today, as people use newer methods for factor depletion and for assaying transcription. Most labs have converged on the view that all or nearly all genes use TFIID, and that SAGA probably also localizes to most genes, but that only roughly 10% show a very strong defect in its absence. Combining TFIID and SAGA defects generally leads to much stronger effects.

This new paper from the Hahn lab, while extremely rigorous and very well done, generally aligns with the Huisinga and Pugh conclusions. A "TFIID/SAGA redundant" class of genes roughly corresponds with what Pugh called "SAGA dominant". The differences here from Pugh and other earlier papers no doubt reflect the specific systems used, whether the assay is for factor occupancy or function, and how one defines the cutoff for calling a gene affected or unaffected. This paper illustrates the problem by reporting big differences between using SAGA deletion or degron alleles (see my comments below). None of the recent genomics papers explains what makes a small fraction of promoters particularly sensitive to SAGA. This paper does rule out that it's the presence of a consensus TATA box, and finds that TFIID/SAGA redundant genes tend to be more highly expressed, but what that means mechanistically is still unclear. At this point, I suspect only specialists are still following this debate.

Major points:

1) The effects of spt3 δ are much stronger effect than Spt3 depletion, and the authors use this to propose different functions of SAGA that manifest over different time scales. There are at two far more likely explanations that need to be considered. First, the weaker effects in the degron mutants could simply be due to incomplete degradation, as residual protein is clearly seen in the western blots. Alternatively, the drop in transcription seen in the deletions could be a long term adjustment to slow growth. To rule out indirect effects, have the authors ever assayed a similarly slow growing yeast mutant in a gene that is not directly involved in transcription?

2) The authors propose that the long-term gene expression effects of SAGA deletions are due to loss of chromatin modifications: in that case it would make sense to test the gcn5 deletion or, even better, an inactive point mutant that can't acetylate. The Gcn5 point mutant grows pretty well, so this could help rule out the indirect growth effect described above. If H2Bub is involved, testing a ubp8 deletion would also be important.

3) Discussion final paragraph. How can SAGA affect "ongoing" transcription but not "activation per se". Does this imply that "ongoing" transcription no longer requires an activator? How would that work? Isn't it simply more likely that the rapid depletion is incomplete?

4) What is the difference between a transcription burst and efficient reinitiation? Isn't a burst defined as a series of closely spaced reinitiations?

---

## [Author Response]

[Editors’ note: The authors successfully appealed and what follows is the authors’ response to the first round of review.]

Reviewer #1:This manuscript is the next installment of a series of papers over several years that have attempted to discern which yeast genes are regulated by the two coactivators SAGA and TFIID. As different techniques have been used, the view has changed. In these current experiments, the authors use metabolic labeling with 4-thioU to measure the level of transcription under either mutant conditions or after depletion of SAGA or TFIID components. Interestingly, they obtain very different results for SAGA between the depletion and deletion conditions. From the depletion studies, they conclude that approximately 13% of the yeast genes that they could assay are regulated in a partially redundant fashion by both SAGA and TFIID, while the remainder are primarily regulated by TFIID. The interpretation is reasonable, although their deletion studies show a much greater effect of SAGA mutants and their ChEC studies, in agreement with previous studies, show that SAGA is present at most promoters, as is TFIID. Overall, this is nicely done work that should be of interest to those in the transcription field. Several comments on both the presentation and the experiments are below.1) The first sentence of the Abstract is inaccurate as SAGA does more than chromatin modifications, as the authors reveal in the Introduction. I suggest revising the sentence to the following: "Deletions of genes encoding subunits of the yeast coactivator SAGA reveal a general role in transcription linked to the multiple activities of this complex."2) Please edit throughout the manuscript to correct little writing errors. As one example, in the Abstract, "…87% of expressed genes we term "TFIID-dependent" are highly sensitive to TFIID depletion…" the word "that" should be inserted before "we term." Also, change "contacts" to "contact." Here's another example: "The genome-wide specificity of SAGA and TFIID have been…" – either change specificity to specificities or change have to has. There are more cases like this throughout the manuscript.3) Introduction section – "relatively weak TBP binding" might be biochemically accurate but it implies a weak activity with respect to the degree of gene regulation, which is not the case for spt3 and spt8 mutants.

Thank you for your comments. The text and figures of the manuscript have been extensively revised to include our new results and to address reviewer comments. The revisions include the suggested wording and other changes listed above in points 1-3 where still relevant in the revised version.

4) Figure 1 and accompanying results – In the comparisons of deletions versus depletions, it would be helpful if the authors clarified their choices for comparison since they are comparing single gene deletions to multiple subunit depletions. Presumably they chose to deplete multiple SAGA subunits at once to increase the chances of efficient SAGA depletion, but please state that explicitly of that is the case. In an ideal world, they would have compared deletion/depletion of the same subunits, such as Spt3 vs Spt3 or Spt3+Spt7 vs Spt3+Spt7. It's formally possible, for example, that depletion of only Spt3 would have a more severe phenotype than the double depletion, although this is unlikely based on previous studies. If they have any unpublished evidence about this, they might mention it.

Now in the revised manuscript: “To increase the probability of efficient SAGA inactivation, double degron strains permitted simultaneous depletion of two SAGA subunits, either Spt3/7 or Spt3/20”.

FYI, our preliminary experiments using Pol II-ChIP assays (not shown) indicated that single SPT deletions had only modest genome-wide transcription defects. Therefore, in subsequent experiments, we transitioned to using the double degron strains to ensure efficient SAGA depletion.

5) Figure 1—figure supplement 4C – What is being correlated in this table? If each experiment was generally done in triplicate, for each genotype, what was used to general the correlation coefficient? Please explain in the legend.

As described in the Materials and methods, the results of biological replicates (two or three) were averaged and the log_2_ change in expression values per gene were calculated for each experiment. These log_2_ change values were used for all relevant plots. The figure legend was changed to provide a more complete description.

6) Since Tafs are essential for viability, what was done to ensure that the cells were still viable during the RNA measurements, for either Taf depletions or Taf + SAGA depletions?

New Figure 1—figure supplement 4B shows the viability of strains after 30 min of IAA treatment. Strains are ≥85% viable after incubation with IAA.

7) Regarding depletion versus deletion, it seems possible that the difference is caused by the level of depletion rather than the time without the protein. Even though the authors have tried to rule this out, I think the formal possibility should be mentioned.

After IAA addition, the levels of most tagged proteins are reduced ~90% but it’s formally possible that the ≥10x depleted levels of intact SAGA are significantly contributing to gene expression. We feel that this is very unlikely for two reasons: (a) The double degron tags have ~10% of each tagged subunit remaining. If each subunit is degraded independently of the other, that means that there is only ~1% of intact SAGA after IAA incubation. (b), The results of the Spt3/Taf13 and Spt7/Taf13 double degron strains show that depletion of only one SAGA subunit is enough to inactivate SAGA (Figure 2B). This result shows that the weak effects of the SAGA degrons in otherwise WT cells is due to substantial redundancy with TFIID function at the coactivator-redundant genes rather than incomplete depletion. A short summary of the above points can be found in the final paragraph of subsection “SAGA has two separate roles in transcription”.

8) Subsection “SAGA-regulated chromatin modifications change little during rapid SAGA inactivation” – For SAGA's role in activation, the authors should also cite three other references – Larschan et al., 2001, Bhaumik et al., 2001, and Dudley et al., 1999.

The references are now included (subsection “SAGA-dependent chromatin modifications change slowly after rapid SAGA depletion”).

9) Figure 4 and subsection “SAGA-regulated chromatin modifications change little during rapid SAGA inactivation” – The authors state that it was unexpected to observe lower H2B-Ub levels in some of the SAGA mutants in panel A. Similarly, it was unexpected to observe lower H3K18-Ac levels in the spt3 deletion strain, based on previous genetic, biochemical, and structural studies that strongly suggest that the HAT module is physically and biochemically distinct from Spt3. Therefore, the same reasoning used for the effect on H2B-UB might apply for the reduction in H3K18-Ac levels in the spt3 mutant. If the authors wish to impose histone modifications as a reason for the depletion vs. deletion difference (Discussion paragraph three), this might be mentioned, as an spt3 deletion has a severe effect on transcription yet in theory it should not affect the level of histone modifications.

Partly because of another reviewer’s suggestion about using ChIP-seq for H3-Ac mapping, the experiments in this section have been entirely redone to include H3K18-Ac ChIP-seq and a more extensive time course of chromatin modification changes upon activation of the Spt-degrons (Figure 5, Figure 5—figure supplements 1,2). The new results are now in line with what is expected for Ubp8 – deletion of this gene has little or no effect on H3K18-Ac. However, the results with the *spt3Δ* strain still hold – this deletion causes a decrease in H2B-Ub levels. In addition, we show that the *spt3Δ* has a defect in H3K18-Ac levels.

Combined with our other results, we think that the new experiments make a strong case that changes in chromatin modifications observed in the *SPT* deletions play a large part in the differences between transcription defects in the deletion vs depletion approaches.

10) Figure 5, subsection “Rapid SAGA depletion primarily affects maintenance of ongoing transcription rather than gene activation by Gcn4” paragraph two, This was an interesting experiment. In contrast to what the authors write, the fold induction looks to be greater in the IAA-treated samples in 5B compared to the non-IAA treated cells in 5A, particularly for ARG3 and HIS4.

The reviewer is correct, and we have revised the manuscript with numbers indicating the fold induction in all cases (subsection “Rapid SAGA depletion primarily affects maintenance of ongoing transcription rather than gene activation by Gcn4”).

11) Figure 1 and 2 legends and probably elsewhere – SC media is stated as the growth medium but it is not described in the Materials and methods. Please clarify.

The Materials and methods have been updated to include the definition of YPD and SC media.

12) The authors need to provide a list of the yeast strains, including names and genotypes.

Yeast strains are listed in Supplementary file 9.

13) The authors discuss the SAGA-TFIID redundancy in the Discussion but they don't offer an explanation for two aspects of the redundancy. First, why is it partial? If either complex can work at those genes, why does expression decrease at all? Is it due to insufficient levels of the other complex? Second, what does the reduced expression represent? Is it caused by normal levels from a subpopulation of cells in the culture, or is it due to reduced expression from all of the cells? In future experiments, this could be addressed by single cell analysis.

These are all great questions. For some reason, the coactivator-dependent subset of genes can’t efficiently utilize only TFIID or only SAGA to attain normal levels of transcription. At the smaller subset of the exceptionally highly-expressed genes, it might make sense that one or the other complex is limiting, but not at the many other genes in this class that are spread throughout the range of expressed genes. We’ve added a paragraph to the last sentence of the Discussion mentioning this issue. Addressing this question promises to be complex and it will be one of the important issues to investigate going forward with this project.

Reviewer #2:In this manuscript, Donczew et al. revisit the highly-debated topic of yeast promoter classes, using acute depletion of TFIID and SAGA subunits and nascent transcript labeling. Older studies by the Pugh lab categorized yeast promoters as SAGA-dominated and TFIID-dominated. In general, the SAGA-dominated genes were described as TATA-enriched and stress responsive, while the TFIID-dominated genes lacked consensus TATA boxes. In a 2017 Molecular Cell paper (Baptista et al. 68:130), Hahn and collaborators showed that deletion of SAGA subunits significantly reduced nascent transcript levels at both TFIID- and SAGA-dominated gene classes and also showed that SAGA occupies the promoters of both gene classes by ChEC-seq. This paper threw into question the classification of promoters by coactivator dependencies and suggested that the functions of SAGA are more widespread than previously thought. However, recently, the ChEC-seq profiles reported in the Baptista paper have been disputed by the Pugh lab in a paper in press at Molecular Cell.After acutely depleting SAGA and TFIID subunits and measuring nascent transcription, the authors in this paper conclude that there are two major classes of Pol II promoters with respect to TFIID and SAGA dependencies: a large TFIID-dependent class and a smaller TFIID/SAGA-redundant class, which accounts for 13% of promoters tested. The authors define these classes through measurements of nascent RNA changes in response to factor depletion using the auxin-inducible degron system, followed by k-means clustering. By co-depleting Taf13 and either Spt3 or Spt7, the authors provide strong support for the redundancy model. They also report results from a new ChEC-seq protocol which show widespread TFIID and SAGA occupancy at all genes in line with the Baptista paper. Regarding the mechanistic distinctions between the TFIID-dependent and TFIID/SAGA-redundant promoters, the authors conclude that gene expression levels do not dictate whether a promoter is redundantly regulated by the coactivators. By reanalyzing the TATA sequences in the promoter classes, the authors report that TFIID/SAGA-redundant promoters are not biased toward TATA-containing versus TATA-less. Western data show that two histone modifications regulated by SAGA do not greatly change during the AID depletion time course. From this the authors conclude that some other function of SAGA, such as a protein-protein interaction, is being disrupted by the depletion and leading changes in transcription. Using RT-PCR, the authors test the effect of SAGA depletion on activation of three Gcn4-regulated genes and find that activation is unaffected, whereas ongoing transcription is reduced.The main conclusion from this paper is the redefining of yeast promoters using acute depletion and nascent transcript labeling. The results support a large TFIID-dependent class and smaller class that can utilize either TFIID or SAGA for their regulation. The molecular basis of the two classes defined here remains uncertain.

Thank you for your comments. One misunderstanding above is that we find the coactivator-redundant gene class is enriched for TATA compared with the TFIID-dependent class. Based on your comments below, we have expanded the original analysis on the difference between the two gene classes in Figure 3 and Supplementary file 5 of the revised manuscript. This additional analysis adds to our prior conclusions as to the mechanistic basis for the difference between the gene classes.

Specific comments:1) The identification of two promoter classes is the main conclusion of the paper and is based on acute protein depletion using the AID system, 4-thioU RNA-seq, and k-means clustering. The paper requires additional controls to show that the AID tags are not disrupting protein function, that auxin or DMSO isn't affecting expression of an untagged WT strain, and that nascent transcripts are enriched (such as by showing browser tracks of intron transcripts). Of note, in Figure 4B, levels of H2B-Ub at the 0 min time point are lower than the WT control. Are the tags affecting the deubiquitination function of SAGA? Details of the clustering analysis are lacking, and a heatmap showing the clusters would be useful. Details on what the authors call an effect on transcription are lacking. Was every gene that passed the filter described in the Materials and methods included in the clustering regardless of the magnitude of the effect on transcription?

Thank you for these comments. Additional controls and experiments are included in the revised manuscript to address these points:

a) Figure 1—figure supplement 3A plots levels of gene expression in WT and all degron strains when DMSO is added. We find no major changes in gene expression due to the degron fusions.

b) Figure 2A, #1, shows that IAA addition causes no transcription changes in WT cells.

c) After purification of labeled mRNA, we routinely test one sample per prep for enrichment of labeled RNA by RT qPCR, probing both total and labeled RNA at 3 transcribed genes. The purified 4TU RNA typically contains 2-10% contamination with unlabeled RNA (now stated in Materials and methods). We also analyzed the data for enrichment of intron-containing RNA in the purified 4TU-labeled samples as requested (Figure 1—figure supplement 3B). Intron-containing RNA is clearly enriched in the purified labeled RNA as expected.

d) The experiment examining degron-mediated chromatin modifications was completely redone to include a longer time course after IAA addition and to examine modifications by ChIP-seq (Figure 5 and Figure 5—figure supplements 1,2). Before IAA addition, the levels of H2B-Ub in WT and degron strains are nearly identical (Figure 5—figure supplement 1B).

e) The revised methods section has additional details on clustering and a heatmap is now included as requested (Figure 1B). As stated in Materials and methods and text, every gene that passed our filters (~83% of all protein-coding genes) was used in clustering. We did not use the magnitude of the effect on transcription as a criteria for clustering as this would invalidate the analysis by significantly limiting the available gene set.

2) The colors highlighting the two classes of promoters are switched in the middle of the paper. This needs to be corrected.

Sorry about that – this has now been corrected.

3) Figures 2, 3 and S3: The axis label "gene expression rank" is confusing. I assume the #1 gene is that with the highest expression but this is not clearly stated.

The figures and legends have been modified to clarify.

4) Figure 3A. The authors conclude that the classification of TFIID/SAGA-redundant genes is not based on gene expression levels. This seems to be an overstatement. If the authors included percentages in panel A, it would seem clear that the TFIID/SAGA-redundant genes represent a larger percentage of the top quintile than the lower quintiles.

There was a misunderstanding on this point. In the original manuscript we explained that, while the coactivator-redundant genes are found in genes at all levels of expression, this gene class is clearly enriched in the highest expressed genes (see Figure 3). This finding is in agreement with the reviewer’s conclusion.

5) Figure 3D. It is unclear how robust the distinction between TATA+ and TATA-less promoters is. Now that the authors have discrete gene sets, why not perform a de novo motif analysis to identify TATA or TATA-like elements as well as other cis-regulatory information that might underlie the coactivator requirements? This type of analysis could provide some mechanistic understanding of the gene classes, which is currently lacking from the field.

Thank you for the suggestion! This analysis is shown in Figure 3D, E and in Supplementary file 5 and this analysis added important new information. While we find that the motifs for 14 transcription factors are enriched in one vs the other gene class, it’s clear that there is no motif that is exclusive for the TFIID-dependent or coactivator-redundant genes. From this, we conclude that the distinction between gene classes is due to multiple components that could include regulatory transcription factors, promoter sequence, and possibly chromatin architecture.

6) Figure 4. How many repetitions were performed and used to calculate the western signals? In panel B, were the lanes on the left spliced from the same blot, exposed for the same length of time? Ubiquitination is unstable and typically extracts are prepared in denaturing buffers. Information on extract preparation is needed.

As explained above, this experiment was completely redone to incorporate additional experimental conditions. All experiments were conducted in biological duplicate. Images in Figure 5—figure supplement 1A are from the same un spliced gel. Images in Figure 5—figure supplement 1B are from two gels (too many samples for one gel), with the normalization controls on each gel shown. Extracts were made in Urea-containing buffers and extract preparation details are now in Materials and methods.

7) Figure 4. From the western analysis, the authors conclude that loss of SAGA-dependent histone modifications cannot explain the changes in transcription that occur in response to SAGA depletion. However, because only 13% of promoters are dependent on SAGA, it is unclear if changes in SAGA-dependent histone modifications would be visible by western. ChIP experiments are needed to test the histone modifications at promoters in each class upon factor depletion.

Thank you for the suggestion. ChIP-seq results for Gcn4-dependent H3K18-Ac are now shown in Figure 5C and Figure 5—figure supplement 2. This, along with the more extensive Western analysis and the results of transcription defects in the *gcn5Δ* and *ubp8Δ* strains (Figure 4), strengthens our conclusions about the role of H3 acetylation causing the differences between SAGA deletions and depletions.

We also attempted ChIP-seq (and cut-and-run) for genome-wide H2B-Ub mapping, but we found that, with our experimental conditions, the available commercial antibodies are not suitable for these methods due to very low numbers of sequencing reads recovered.

8) Figure 5. More experimental details are needed.

We have carefully gone over the text, figure legends and methods and feel that the experiment is clearly described. If something specific is not clear, please let us know.

9) Figure 6. Except for changes in color, both of these panels are the same as those included in the Donczew et al. paper on modified ChEC-seq that the authors cite as submitted for publication and was provided to the reviewers of this paper. This is clearly unacceptable. Frankly, I feel that the authors should include the modified ChEC-seq protocol, analysis, and results in this paper and not in a separate paper.

There was a misunderstanding about what Figure 6 represented. Our letter to Mol Cell in response to Pugh et al. showed ChEC-seq mapping of TFIID and SAGA to genes in relation to the Pugh et al. “TFIID-dominated” and “SAGA-dominated” gene categories. Figure 6 in the original *eLife* submission showed how SAGA and TFIID map to the newly defined gene categories “TFIID-dependent” and “coactivator-redundant”. In any case, since Mol Cell has not yet made a decision on our letter, we included the binding data in this revised manuscript as requested and removed it from a revised Mol Cell communication. This binding data is now shown in Figure 6 and Figure 6—figure supplement 1 of the revised *eLife* manuscript.

10) Figure 1—figure supplement 4C. Correlations between data sets are more standardly shown as xy plots. This graph is confusing. What does "genes left" mean?

This is now explained in the Figure legend (Figure 1—figure supplement 4C).

11) Figure 3—figure supplement 1A. A log scale on the y-axis would be better.

We replotted this graph, illustrating the difference between Pugh’s original gene categories and our new ones, on a log scale. However, my coauthors and I felt that this gave a misleading first impression that the differences between the new and old gene categories are huge. Given this, we prefer to leave this plot as a linear scale unless you feel strongly otherwise.

Reviewer #3:It's hard to believe we're still arguing about how many and which kinds of genes use TFIID and/or SAGA. This debate goes back to the mid-1990s, when Struhl and Green used TS alleles to question whether TAFs really had a role in activation. In 2004, the Pugh lab classified about 10% of genes as "SAGA-dominant" and the rest as "TFIID-dominant". The arguments continue today, as people use newer methods for factor depletion and for assaying transcription. Most labs have converged on the view that all or nearly all genes use TFIID, and that SAGA probably also localizes to most genes, but that only roughly 10% show a very strong defect in its absence. Combining TFIID and SAGA defects generally leads to much stronger effects.This new paper from the Hahn lab, while extremely rigorous and very well done, generally aligns with the Huisinga and Pugh conclusions. A "TFIID/SAGA redundant" class of genes roughly corresponds with what Pugh called "SAGA dominant". The differences here from Pugh and other earlier papers no doubt reflect the specific systems used, whether the assay is for factor occupancy or function, and how one defines the cutoff for calling a gene affected or unaffected. This paper illustrates the problem by reporting big differences between using SAGA deletion or degron alleles (see my comments below). None of the recent genomics papers explains what makes a small fraction of promoters particularly sensitive to SAGA. This paper does rule out that it's the presence of a consensus TATA box, and finds that TFIID/SAGA redundant genes tend to be more highly expressed, but what that means mechanistically is still unclear. At this point, I suspect only specialists are still following this debate.

Thank you for your comments, however, we respectfully disagree with many of the above statements. I think that some of the above comments give an inaccurate picture on the current state of the field and how our work is significantly different from the prior consensus. I believe that the new findings reported in our manuscript will have a significant and lasting impact on understanding the function and gene-specificity of TFIID and SAGA.

1) We strongly disagree with the statement that “Most labs have converged on the view that all or nearly all genes use TFIID…” In my experience (based on recently published papers and meeting presentations), most labs are using the terminology “TFIID-dominated” (or Taf-dependent) and SAGA-dominated (or Taf-independent). A recent example of this thinking is a 2019 *eLife* paper by Struhl and colleagues (PMID: 30681409), which is clearly different from the reviewer’s characterization that “all or nearly all genes use TFIID”. One of the Petrenko et al. conclusions was that many of the most highly expressed genes show minimal use of TFIID. Another recent example is in Randy Morse’s *eLife* paper (Knoll et al., 2018, PMID: 30540252) where the results of gene-specific Mediator mutations are interpreted in terms of “SAGA-dominated” and “TFIID-dominated genes”.

2) The reviewer goes on to state: “Most labs have converged on the view that ……. SAGA probably also localizes to most genes, but that only roughly 10% show a very strong defect in its absence.” Based on the literature and recent meeting presentations, a widely held view is that SAGA localizes most strongly to the “SAGA” genes and TFIID localizes most strongly to the “TFIID” genes. I believe that our genome-wide results on finding that SAGA and TFIID bind to many genes of both gene classes is not yet (but will soon be) a generally accepted view.

In contrast to the above view “that only roughly 10% [of genes] show a very strong defect in [SAGAs] absence”, our new work shows that transcription from nearly all genes is decreased ≥2-fold in the spt3Δ and spt7Δ strains. It’s only when SAGA is rapidly depleted that we observe modest gene-specific transcription defects at ~13% of genes.

3) The reviewer states that: “Combining TFIID and SAGA defects generally leads to much stronger effects.” This view, quoted in many papers, is primarily based on Figure 1 in Huisinga and Pugh, 2004; PMID: 14992726. In brief, the basis for this widely quoted statement is not consistent with our recent findings and was likely based in part on measuring steady state RNA instead of newly-synthesized RNA and the technical limitations of RNA microarray assays.

(4) We strongly disagree with this general characterization about alignment of our results with the Huisinga et al. conclusions: “This new paper from the Hahn lab, while extremely rigorous and very well done, generally aligns with the Huisinga and Pugh conclusions”. Our manuscript shows that, (a) rather than expression being “dominated” by reliance on one or the other coactivator, there is an important class of genes that can rely on either TFIID or SAGA for their regulation, (b) that the overlapping acute function of the two coactivators is confined to a specific gene set, not all genes, and (c) that the set of the coactivator-redundant genes is >50% different from the Huisinga et al. gene set (our Figure 3—figure supplement 1). This isn’t a trivial difference. As explained in the manuscript, some prior findings in the literature were incorrectly interpreted based on the now outdated gene categories.

Major points:1) The effects of spt3 δ are much stronger effect than Spt3 depletion, and the authors use this to propose different functions of SAGA that manifest over different time scales. There are at two far more likely explanations that need to be considered. First, the weaker effects in the degron mutants could simply be due to incomplete degradation, as residual protein is clearly seen in the western blots. Alternatively, the drop in transcription seen in the deletions could be a long term adjustment to slow growth. To rule out indirect effects, have the authors ever assayed a similarly slow growing yeast mutant in a gene that is not directly involved in transcription?

A) To rule out indirect effects of the degron tag on SAGA and TFIID subunits and SAGA deletion experiments, we compared our transcription results with the stress-response transcription signature derived from transcriptome profiles of many mutant yeast strains (O’Duibhir et al., 2014). For all strains assayed in our work, the changes in gene expression showed poor correlation with this stress-response signature, indicating that the changes we observed are not due to a general stress response. Please see paragraph three subsection “SAGA has two separate roles in transcription” and Figure 1—figure supplement 4A.

B) After IAA addition, the levels of most tagged proteins are reduced ~90% but it’s formally possible that the 10x lower levels of intact SAGA are significantly contributing to gene expression. We feel that this is very unlikely for two reasons: (a) The double degron tags have ~10% of each tagged subunit remaining. If each subunit is degraded independently of the other, that means that there is only ~ 1% of intact SAGA after IAA incubation. (b), The results of the Spt3/Taf13 and Spt7/Taf13 double degron strains shows that depletion of only one SAGA subunit is enough to inactivate SAGA (Figure 2B). This figure shows that the weak effects of the SAGA degrons is due to substantial redundancy with TFIID function at the coactivator-redundant genes rather than from incomplete depletion. A short summary of these points can be found in paragraph six of subsection “SAGA has two separate roles in transcription”.

2) The authors propose that the long-term gene expression effects of SAGA deletions are due to loss of chromatin modifications: in that case it would make sense to test the gcn5 deletion or, even better, an inactive point mutant that can't acetylate. The Gcn5 point mutant grows pretty well, so this could help rule out the indirect growth effect described above. If H2Bub is involved, testing a ubp8 deletion would also be important.

Thank you for the suggestion! We did 4thioU RNA-seq with *gcn5Δ* and *ubp8Δ* strains in combination with a longer time course of degron-depletion and H3K18-Ac ChIP-seq (Figures 4-5; Figure 5—figure supplements 1,2). These additional experiments make a strong case for the role of SAGA-dependent chromatin modifications playing a large part in the different transcription phenotypes of *SPT* deletions and *SPT* depletion by degron.

3) Discussion final paragraph. How can SAGA affect "ongoing" transcription but not "activation per se". Does this imply that "ongoing" transcription no longer requires an activator? How would that work? Isn't it simply more likely that the rapid depletion is incomplete?4) What is the difference between a transcription burst and efficient reinitiation? Isn't a burst defined as a series of closely spaced reinitiations?

Thank you for pointing out these confusing points. The last section in the Discussion has been rewritten to correct and clarify.